# Learning cortical representations through perturbed and adversarial dreaming

**Nicolas Deperrois[1]\*, Mihai A Petrovici[1,2], Walter Senn[1†], Jakob Jordan[1†]**

[1]Department of Physiology, University of Bern, Bern, Switzerland; [2]Kirchhoff-Institute for Physics, Heidelberg University, Heidelberg, Germany

**Abstract** Humans and other animals learn to extract general concepts from sensory experience without extensive teaching. This ability is thought to be facilitated by offline states like sleep where previous experiences are systemically replayed. However, the characteristic creative nature of dreams suggests that learning semantic representations may go beyond merely replaying previous experiences. We support this hypothesis by implementing a cortical architecture inspired by generative adversarial networks (GANs). Learning in our model is organized across three different global brain states mimicking wakefulness, non-rapid eye movement (NREM), and REM sleep, optimizing different, but complementary, objective functions. We train the model on standard datasets of natural images and evaluate the quality of the learned representations. Our results suggest that generating new, virtual sensory inputs via adversarial dreaming during REM sleep is essential for extracting semantic concepts, while replaying episodic memories via perturbed dreaming during NREM sleep improves the robustness of latent representations. The model provides a new computational perspective on sleep states, memory replay, and dreams, and suggests a cortical implementation of GANs.

**\*For correspondence:**
nicolas.deperrois@unibe.ch

[†]These authors shared senior authorship to this work

**Competing interest:** The authors declare that no competing interests exist.

## Editor's evaluation

This paper presents a generative adversarial network-inspired model of how learning during wakefulness, non-rapid eye movement (NREM), and REM sleep work together to facilitate the emergence of object category representations. The model is impressive in its ability to shape representations based on internally generated activity that does not directly recapitulate prior experience, and has properties that correspond to replay and dreams in NREM and REM sleep. The model makes predictions that can be tested in sleep experiments in humans and animals.

## Introduction

After just a single night of bad sleep, we are acutely aware of the importance of sleep for orderly body and brain function. In fact, it has become clear that sleep serves multiple crucial physiological functions (*Siegel, 2009*; *Xie et al., 2013*), and growing evidence highlights its impact on cognitive processes (*Walker, 2009*). Yet, a lot remains unknown about the precise contribution of sleep, and in particular dreams, on normal brain function.

One remarkable cognitive ability of humans and other animals lies in the extraction of general concepts and statistical regularities from sensory experience without extensive teaching (*Bergelson and Swingley, 2012*). Such regularities in the sensorium are reflected on the neuronal level in invariant object-specific representations in high-level areas of the visual cortex (*Grill-Spector et al., 2001*; *Hung et al., 2005*; *DiCarlo et al., 2012*) on which downstreams areas can operate. These so-called semantic representations are progressively constructed and enriched over an organism's lifetime

(*Tenenbaum et al., 2011*; *Yee et al., 2013*), and their emergence is hypothesized to be facilitated by offline states such as sleep (*Dudai et al., 2015*).

Previously, several cortical models have been proposed to explain how offline states could contribute to the emergence of high-level, semantic representations. Stochastic hierarchical models that learn to maximize the likelihood of observed data under a generative model such as the Helmholtz machine (*Dayan et al., 1995*) and the closely related Wake–Sleep algorithm (*Hinton et al., 1995*; *Bornschein and Bengio, 2015*) have demonstrated the potential of combining online and offline states to learn semantic representations. However, these models do not leverage offline states to improve their generative model but are explicitly trained to reproduce sensory inputs during wakefulness. In contrast, most dreams during REM sleep exhibit realistic imagery beyond past sensory experience (*Fosse et al., 2003*; *Nir and Tononi, 2010*; *Wamsley, 2014*), suggesting learning principles that go beyond mere reconstructions.

In parallel, cognitive models inspired by psychological studies of sleep proposed a 'trace transformation theory' where semantic knowledge is actively extracted in the cortex from replayed hippocampal episodic memories (*Nadel and Moscovitch, 1997*; *Winocur et al., 2010*; *Lewis and Durrant, 2011*). However, these models lack a mechanistic implementation compatible with cortical structures and only consider the replay of waking activity during sleep.

Recently, implicit generative models that do not explicitly try to reconstruct observed sensory inputs, and in particular generative adversarial networks (GANs; *Goodfellow et al., 2014*), have been successfully applied in machine learning to generate new but realistic data from random patterns. This ability has been shown to be accompanied by the learning of disentangled and semantically meaningful representations (*Radford et al., 2015*; *Donahue et al., 2016*; *Liu et al., 2021*). They thus may provide computational principles for learning cortical semantic representations during offline states by generating previously unobserved sensory content as reported from dream experiences.

Most dreams experienced during rapid eye movement (REM) sleep only incorporate fragments of previous waking experience, often intermingled with past memories (*Schwartz, 2003*). Surprisingly, such random combinations of memory fragments often result in visual experiences that are perceived as highly structured and realistic by the dreamer. The striking similarity between the inner world of dreams and the external world of wakefulness suggests that the brain actively creates novel experiences by rearranging stored episodic patterns in a meaningful manner (*Nir and Tononi, 2010*). A few hypothetical functions were attributed to this phenomenon, such as enhancing creative problem solving by building novel associations between unrelated memory elements (*Cai et al., 2009*; *Llewellyn, 2016*; *Lewis et al., 2018*), forming internal prospective codes oriented toward future waking experiences (*Llewellyn, 2015*), or refining a generative model by minimizing its complexity and improving generalization (*Hobson et al., 2014*; *Hoel, 2021*). However, these theories do not consider the role of dreams for a more basic function, such as the formation of semantic cortical representations.

Here, we propose that dreams, and in particular their creative combination of episodic memories, play an essential role in forming semantic representations over the course of development. The formation of representations that abstract away redundant information from sensory input and that can thus be easily used by downstream areas is an important basis for memory semantization. To support this hypothesis, we introduce a new, functional model of cortical representation learning. The central ingredient of our model is a creative generative process via feedback from higher to lower cortical areas that mimics dreaming during REM sleep. This generative process is trained to produce a more realistic virtual sensory experience in an adversarial fashion by trying to fool an internal mechanism distinguishing low-level activities between wakefulness and REM sleep. Intuitively, generating new but realistic sensory experiences, instead of merely reconstructing previous observations, requires the brain to understand the composition of its sensorium. In line with transformation theories, this suggests that cortical representations should carry semantic, decontextualized gist information.

We implement this model in a cortical architecture with hierarchically organized forward and backward pathways, loosely inspired by GANs. The connectivity of the model is adapted by gradient-based synaptic plasticity, optimizing different, but complementary objective functions depending on the brain's global state. During wakefulness, the model learns to recognize that low-level activity is externally driven, stores high-level representations in the hippocampus, and tries to predict low-level from high-level activity (*Figure 1a*). During NREM sleep, the model learns to reconstruct replayed high-level activity patterns from generated low-level activity, perturbed by virtual occlusions, referred

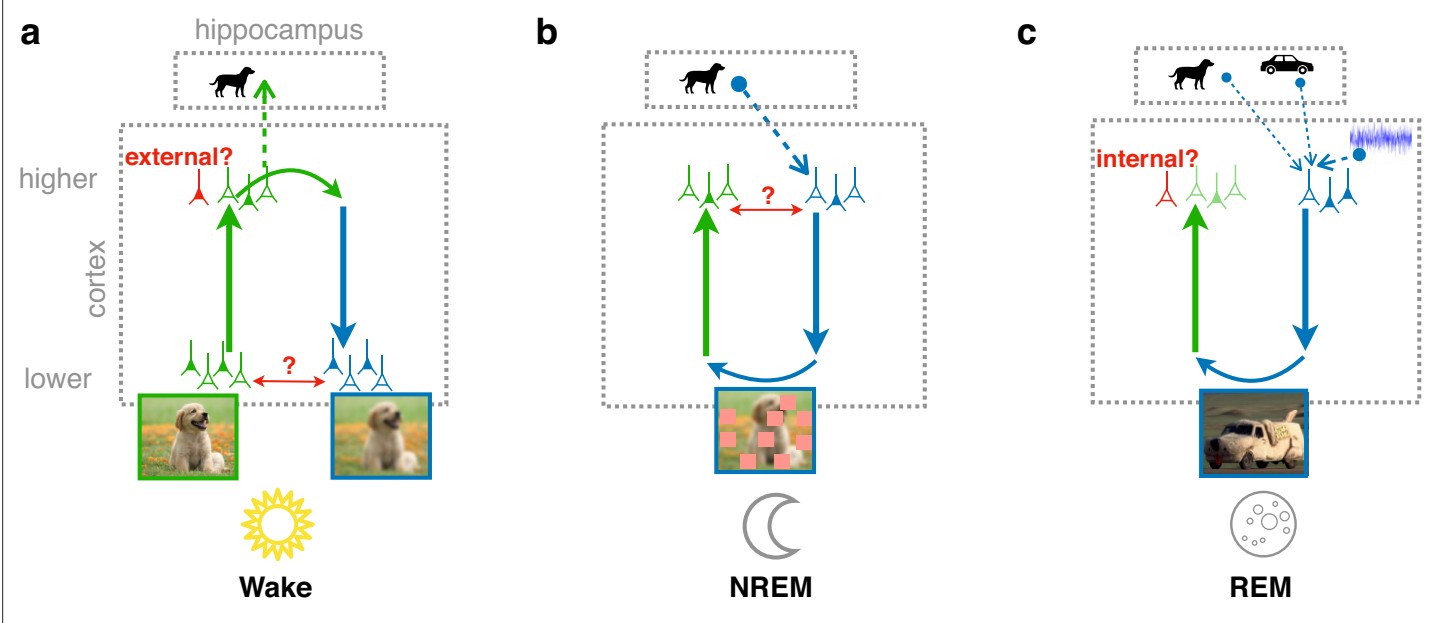

**Figure 1.** Cortical representation learning through perturbed and adversarial dreaming (PAD). (**a**) During wakefulness (Wake), cortical feedforward pathways learn to recognize that low-level activity is externally driven and feedback pathways learn to reconstruct it from high-level neuronal representations. These high-level representations are stored in the hippocampus. (**b**) During non-rapid eye movement sleep (NREM), feedforward pathways learn to reconstruct high-level activity patterns replayed from the hippocampus affected by low-level perturbations, referred to as perturbed dreaming. (**c**) During rapid eye movement sleep (REM), feedforward and feedback pathways operate in an adversarial fashion, referred to as adversarial dreaming. Feedback pathways generate virtual low-level activity from combinations of multiple hippocampal memories and spontaneous cortical activity. While feedforward pathways learn to recognize low-level activity patterns as internally generated, feedback pathways learn to fool feedforward pathways.

to as perturbed dreaming (*Figure 1b*). During REM sleep, the model learns to generate realistic low-level activity patterns from random combinations of several hippocampal memories and spontaneous cortical activity, while simultaneously learning to distinguish these virtual experiences from externally driven waking experiences, referred to as adversarial dreaming (*Figure 1c*). Together with the wakefulness, the two sleep states, NREM and REM, jointly implement our model of perturbed and adversarial dreaming (PAD).

Over the course of learning, constrained by its architecture and the prior distribution of latent activities, our cortical model trained on natural images develops rich latent representations along with the capacity to generate plausible early sensory activities. We demonstrate that adversarial dreaming during REM sleep is essential for learning representations organized according to object semantics, which are improved and robustified by perturbed dreaming during NREM sleep. Together, our results demonstrate a potential role of dreams and suggest complementary functions of REM and NREM sleep in cortical representation learning.

## Results

### Complementary objectives for wakefulness, NREM, and REM sleep

We consider an abstract model of the visual ventral pathway consisting of multiple, hierarchically organized cortical areas, with a feedforward pathway, or encoder, transforming neuronal activities from lower to higher areas (*Figure 2*, $E$). These high-level activities are compressed representations of low-level activities and are called latent representations, here denoted by $z$. In addition to this feedforward pathway, we similarly model a feedback pathway, or generator, projecting from higher to lower areas (*Figure 2*, $G$). These two pathways are supported by a simple hippocampal module that can store and replay latent representations. Three different global brain states are considered: wakefulness (Wake), non-REM sleep (NREM), and REM sleep (REM). We focus on the functional role of these phases while abstracting away dynamic features such as bursts, spindles, or slow waves (*Léger et al., 2018*), in

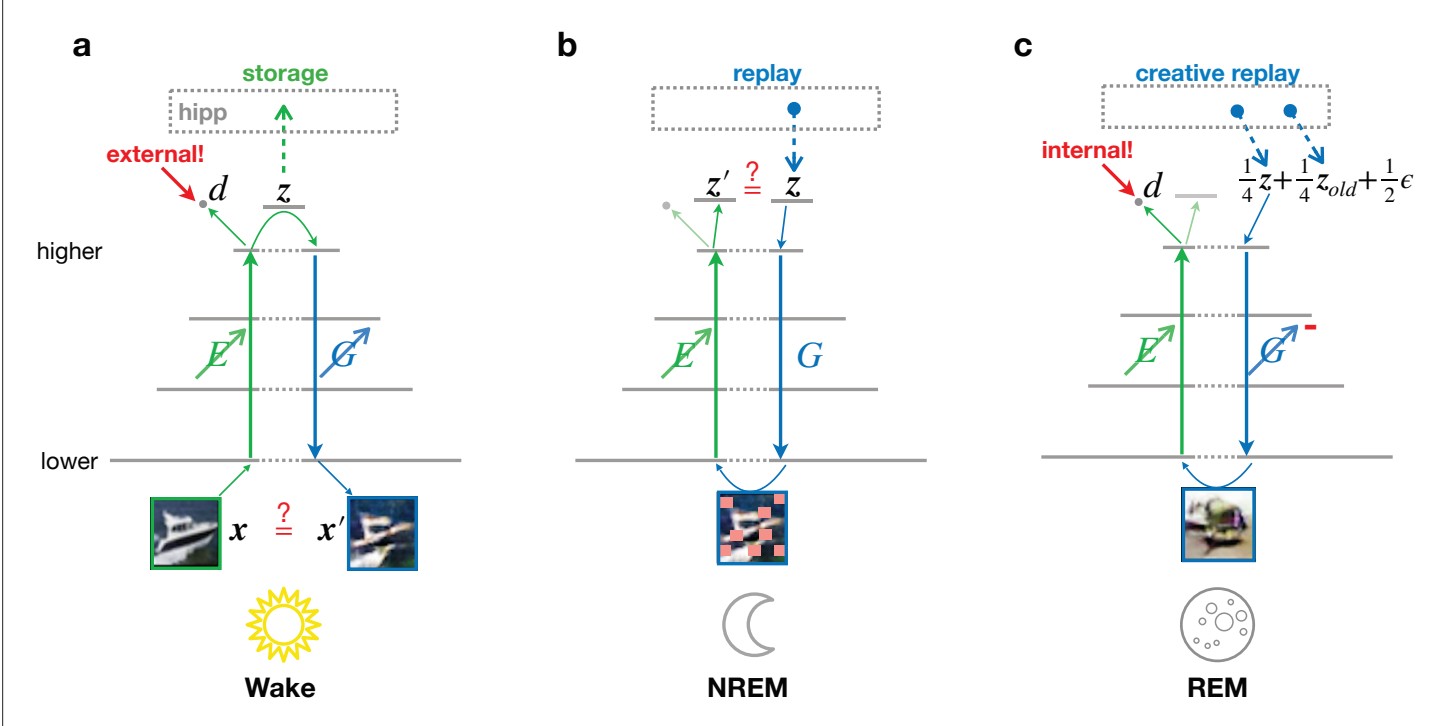

**Figure 2.** Different objectives during wakefulness, non-rapid eye movement (NREM), and rapid eye movement (REM) sleep govern the organization of feedforward and feedback pathways in perturbed and adversarial dreaming (PAD). The variable $x$ corresponds to 32 × 32 image, $z$ is a 256-dimensional vector representing the latent layer (higher sensory cortex). Encoder ($E$, green) and generator ($G$, blue) networks project bottom-up and top-down signals between lower and higher sensory areas. An oblique arrow (↗) indicates that learning occurs in a given pathway. (**a**) During Wake, low-level activities $x$ are reconstructed. At the same time, $E$ learns to classify low-level activity as external (red target 'external!') with its output discriminator $d$. The obtained latent representations $z$ are stored in the hippocampus. (**b**) During NREM, the activity $z$ stored during wakefulness is replayed from the hippocampal memory and regenerates visual input from the previous day perturbed by occlusions, modeled by squares of various sizes applied along the generated low-level activity with a certain probability (see Materials and methods). In this phase, $E$ adapts to reproduce the replayed latent activity. (**c**) During REM, convex combinations of multiple random hippocampal memories ($z$ and $z_{old}$) and spontaneous cortical activity ($\epsilon$), here with specific prefactors, generate a virtual activity in lower areas. While the encoder learns to classify this activity as internal (red target 'internal!'), the generator adversarially learns to generate visual inputs that would be classified as external. The red minus on $G$ indicates the inverted plasticity implementing this adversarial training.

line with previous approaches based on goal-driven modeling that successfully predict physiological features along the ventral stream (*Yamins et al., 2014*; *Zhuang et al., 2021*).

In our model, the three brain states only differ in their objective function and the presence or absence of external input. Synaptic plasticity performs stochastic gradient descent on state-specific objective functions via error backpropagation (*LeCun et al., 2015*). We assume that efficient credit assignment is realized in the cortex and focus on the functional consequences of our specific architecture. For potential implementations of biophysically plausible backpropagation in cortical circuits, we refer to previous work (e.g., *Whittington and Bogacz, 2019*; *Lillicrap et al., 2020*).

During Wake (*Figure 2a*), sensory inputs evoke activities $x$ in the lower sensory cortex that are transformed via the feedforward pathway $E$ into latent representations $z$ in the higher sensory cortex. The hippocampal module stores these latent representations, mimicking the formation of episodic memories. Simultaneously, the feedback pathway $G$ generates low-level activities $x'$ from these representations. Synaptic plasticity adapts the encoding and generative pathways ($E$ and $G$) to minimize the mismatch between externally driven and internally generated activities (*Figure 2a*). Thus, the network learns to reproduce low-level activity from abstract high-level representations. Simultaneously, $E$ also acts as a 'discriminator' with output $d$ that is trained to become active, reflecting that the low-level activity was driven by an external stimuli. The discriminator learning during Wake is essential to drive adversarial learning during REM. Note that computationally the classification of low-level cortical activities into 'externally driven' and 'internally generated' is not

different from classification into, for example, different object categories, even though conceptually they serve different purposes. The dual use of $E$ reflects a view of cortical information processing in which several network functions are preferentially shared among a single network mimicking the ventral visual stream (*DiCarlo et al., 2012*). This approach has been previously successfully employed in machine learning models (*Huang et al., 2018*; *Brock et al., 2017*; *Ulyanov et al., 2017*; *Munjal et al., 2020*; *Bang et al., 2020*).

For the subsequent sleep phases, the system is disconnected from the external environment, and activity in the lower sensory cortex is driven by top-down signals originating from higher areas, as previously suggested (*Nir and Tononi, 2010*; *Aru et al., 2020*). During NREM (*Figure 2b*), latent representations $z$ are recalled from the hippocampal module, corresponding to the replay of episodic memories. These representations generate low-level activities that are perturbed by suppressing early sensory neurons, modeling the observed differences between replayed and waking activities (*Ji and Wilson, 2007*). The encoder reconstructs latent representations from these activity patterns, and synaptic plasticity adjusts the feedforward pathway to make the latent representation of the perturbed generated activity similar to the original episodic memory. This process defines perturbed dreaming.

During REM (*Figure 2c*), sleep is characterized by creative dreams generating realistic virtual sensory experiences out of the combination of episodic memories (*Fosse et al., 2003*; *Lewis et al., 2018*). In PAD, multiple random episodic memories from the hippocampal module are linearly combined and projected to the cortex. Reflecting the decreased coupling (*Wierzynski et al., 2009*; *Lewis et al., 2018*) between hippocampus and cortex during REM sleep, these mixed representations are diluted with spontaneous cortical activity, here abstracted as Gaussian noise with zero mean and unit variance. From this new high-level cortical representation, activity in the lower sensory cortex is generated and finally passed through the feedforward pathway. Synaptic plasticity adjusts feedforward connections $E$ to silence the activity of the discriminator output as it should learn to distinguish it from externally evoked sensory activity. Simultaneously, feedback connections are adjusted adversarially to generate activity patterns that appear externally driven and thereby trick the discriminator into believing that the low-level activity was externally driven. This is achieved by inverting the sign of the errors that determine synaptic weight changes in the generative network. This process defines adversarial dreaming.

The functional differences between our proposed NREM and REM sleep phases are motivated by experimental data describing a reactivation of hippocampal memories during NREM sleep and the occurrence of creative dreams during REM sleep. In particular, hippocampal replay has been reported during NREM sleep within sharp-wave-ripples (*O'Neill et al., 2010*), also observed in the visual cortex (*Ji and Wilson, 2007*), which resembles activity from wakefulness. Our REM sleep phase is built upon cognitive theories of REM dreams (*Llewellyn, 2015*; *Lewis et al., 2018*) postulating that they emerge from random combinations between episodic memory elements, sometimes remote from each other, which appear realistic for the dreamer. This random coactivation could be caused by theta oscillations in the hippocampus during REM sleep (*Buzsáki, 2002*). The addition of cortical noise is motivated by experimental work showing reduced correlations between hippocampal and cortical activity during REM sleep (*Wierzynski et al., 2009*), and the occurrence of ponto-geniculo-occipital (PGO) waves (*Nelson et al., 1983*) in the visual cortex often associated with the generation of novel visual imagery in dreams (*Hobson et al., 2000*; *Hobson et al., 2014*). Furthermore, the cortical contribution in REM dreaming is supported by experimental evidence that dreaming still occurs with hippocampal damage, while reported to be less episodic-like in nature (*Spanò et al., 2020*).

Within our suggested framework, 'dreams' arise as early sensory activity that is internally generated via feedback pathways during offline states, and subsequently processed by feedforward pathways. In particular, this implies that besides REM dreams, NREM dreams exist. However, in contrast to REM dreams, which are significantly different from waking experiences (*Fosse et al., 2003*), our model implies that NREM dreams are more similar to waking experiences since they are driven by single episodic memories, in contrast to REM dreams that are generated from a mixture of episodic memories. Furthermore, the implementation of adversarial dreaming requires an internal representation of whether early sensory activity is externally or internally generated, that is, a distinction whether a sensory experience is real or imagined.

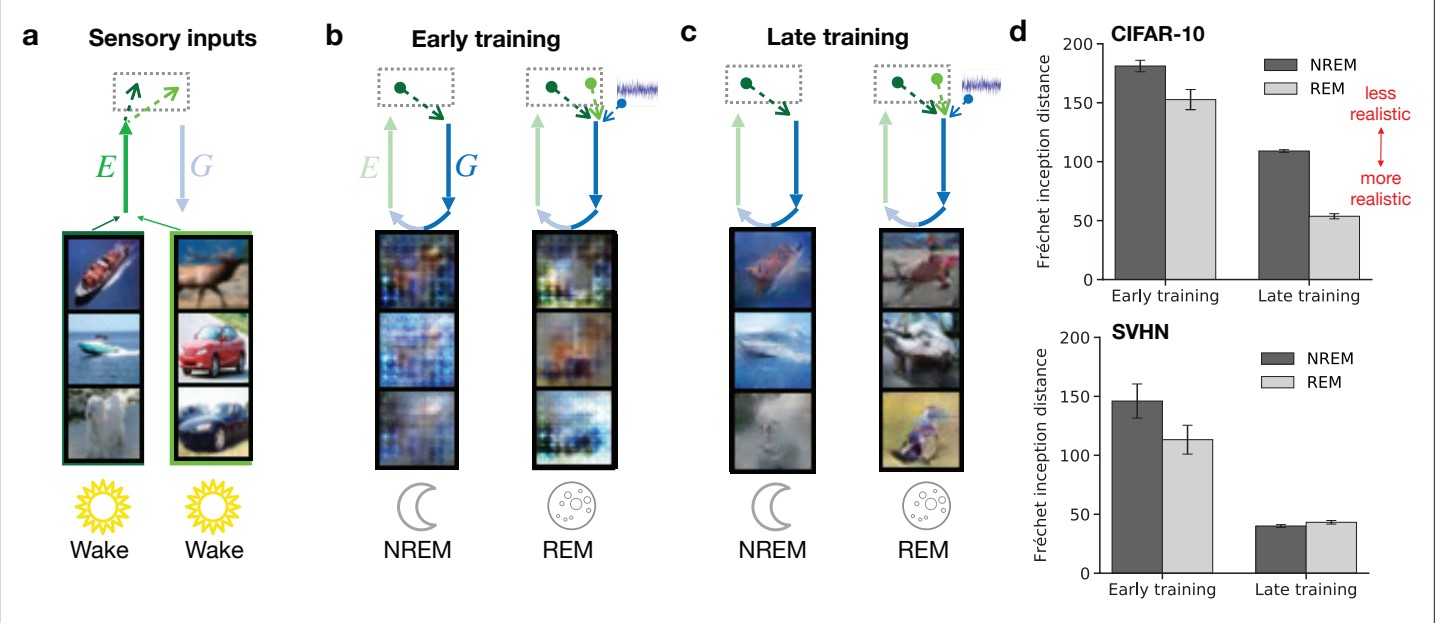

**Figure 3.** Both non-rapid eye movement (NREM) and rapid eye movement (REM) dreams become more realistic over the course of learning. (**a**) Examples of sensory inputs observed during wakefulness. Their corresponding latent representations are stored in the hippocampus. (**b, c**) Single episodic memories (latent representations of stimuli) during NREM from the previous day and combinations of episodic memories from the two previous days during REM are recalled from hippocampus and generate early sensory activity via feedback pathways. This activity is shown for early (epoch 1) and late (epoch 50) training stages of the model. (**d**) Discrepancy between externally driven and internally generated early sensory activity as measured by the Fréchet inception distance (FID) (*Heusel et al., 2018*) during NREM and REM for networks trained on CIFAR-10 (top) and SVHN (bottom). Lower distance reflects higher similarity between sensory-evoked and generated activity. Error bars indicate ±1 SEM over four different initial conditions.

## Dreams become more realistic over the course of learning

Dreams in our model arise from both NREM (perturbed dreaming) and REM (adversarial dreaming) phases. In both cases, they are characterized by activity in early sensory areas generated via feedback pathways. To illustrate learning in PAD, we consider these low-level activities during NREM and during REM for a model with little learning experience ('early training') and a model that has experienced many wake–sleep cycles ('late training'; *Figure 3*). A single wake–sleep cycle consists of Wake, NREM, and REM phases. As an example, we train our model on a dataset of natural images (CIFAR-10; *Krizhevsky and Hinton, 2009*) and a dataset of images of house numbers (SVHN; *Netzer et al., 2011*). Initially, internally generated low-level activities during sleep do not share significant similarities with sensory-evoked activities from Wake (*Figure 3a*); for example, no obvious object shapes are represented (*Figure 3b*). After plasticity has organized network connectivity over many wake–sleep cycles (50 training epochs), low-level internally generated activity patterns resemble sensory-evoked activity (*Figure 3c*). NREM-generated activities reflect the sensory content of the episodic memory (sensory input from the previous day). REM-generated activities are different from the sensory activities corresponding to the original episodic memories underlying them as they recombine features of sensory activities from the two previous days, but still exhibit a realistic structure. This increase in similarity between externally driven and internally generated low-level activity patterns is also reflected in a decreasing Fréchet inception distance (*Figure 3d*), a metric used to quantify the realism of generated images (*Heusel et al., 2018*). The increase of dreams realism, here mostly driven by a combination of reconstruction learning (Wake) and adversarial learning (Wake and REM), correlates with the development of dreams in children, which are initially plain and fail to represent objects, people, but become more realistic and structured over time (*Foulkes, 1999*; *Nir and Tononi, 2010*).

The PAD training paradigm hence leads to internally generated low-level activity patterns that become more difficult to discern from externally driven activities, whether they originate from single episodic memories during NREM or from noisy random combinations thereof during REM. We will next demonstrate that the same learning process leads to the emergence of robust semantic representations.

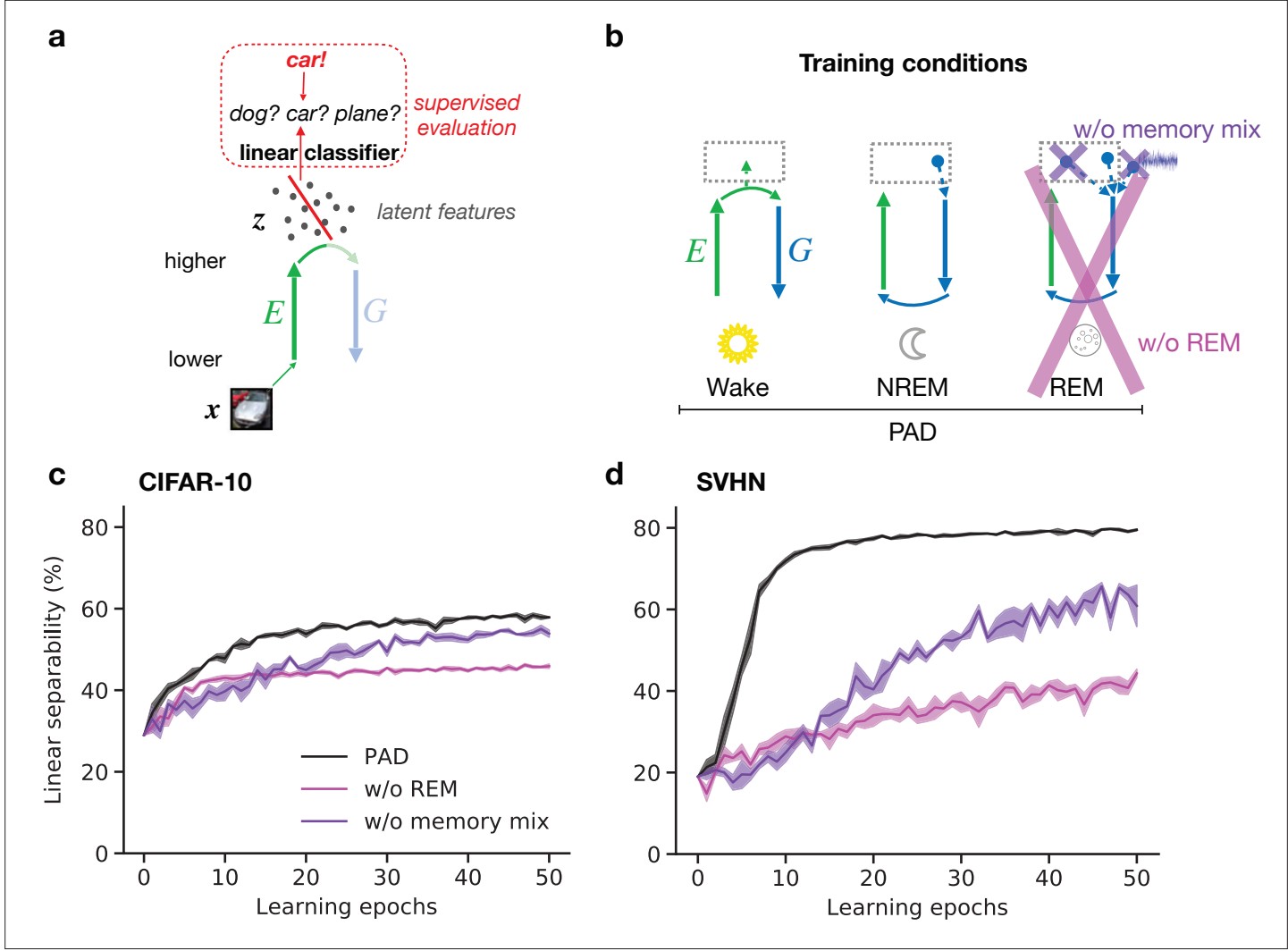

**Figure 4.** Adversarial dreaming during rapid eye movement (REM) improves the linear separability of the latent representation. (**a**) A linear classifier is trained on the latent representations $z$ inferred from an external input $x$ to predict its associated label (here, the category 'car'). (**b**) Training phases and pathological conditions: full model (perturbed and adversarial dreaming [PAD], black), no REM phase (pink) and PAD with a REM phase using a single episodic memory only ('w/o memory mix', purple). (**c, d**) Classification accuracy obtained on test datasets (**c**: CIFAR-10; **d**: SVHN) after training the linear classifier to convergence on the latent space $z$ for each epoch of the $E$-$G$-network learning. Full model (PAD): black line; without REM: pink line; with REM, but without memory mix: purple line. Solid lines represent mean, and shaded areas indicate ±1 SEM over four different initial conditions.

## Adversarial dreaming during REM facilitates the emergence of semantic representations

Semantic knowledge is fundamental for animals to learn quickly, adapt to new environments and communicate, and is hypothesized to be held by so-called semantic representations in the cortex (*DiCarlo et al., 2012*). An example of such semantic representations are neurons from higher visual areas that contain linearly separable information about object category, invariant to other factors of variation, such as background, orientation or pose (*Grill-Spector et al., 2001*; *Hung et al., 2005*; *Majaj et al., 2015*).

Here, we demonstrate that PAD, due to the specific combination of plasticity mechanisms during Wake, NREM, and REM, develops such semantic representations in higher visual areas. Similarly as in the previous section, we train our model on the CIFAR-10 and SVHN datasets. To quantify the quality of inferred latent representations, we measure how easily downstream neurons can read out object identity from these. For a simple linear readout, its classification accuracy reflects the linear separability of different contents represented in a given dataset. Technically, we train a linear classifier that

distinguishes object categories based on their latent representations $z$ after different numbers of wake–sleep cycles ('epochs', *Figure 4a*) and report its accuracy on data not used during training of the model and classifier ('test data'). While training the classifier, the connectivity of the network ($E$ and $G$) is fixed.

The latent representation ($z$) emerging from the trained network (*Figure 4b*, PAD) shows increasing linear separability reaching around 59% test accuracy on CIFAR-10 (*Figure 4c*, black line; for details, see *Appendix 1—table 1*) and 79% on SVHN (*Figure 4d*, black line), comparable to less biologically plausible machine learning models (*Berthelot et al., 2018*). These results show the ability of PAD to discover semantic concepts across wake–sleep cycles in an unsupervised fashion.

Within our computational framework, we can easily consider sleep pathologies by directly interfering with the sleep phases. To highlight the importance of REM in learning semantic representations, we consider a reduced model in which the REM phase with adversarial dreaming is suppressed and only perturbed dreaming during NREM remains (*Figure 4b*, pink cross). Without REM sleep, linear separability increases much slower and even after a large number of epochs remains significantly below the PAD (see also *Appendix 1—figure 3c and d*). This suggests that adversarial dreaming during REM, here modeled by an adversarial game between feedforward and feedback pathways, is essential for the emergence of easily readable, semantic representations in the cortex. From a computational point of view, this result is in line with previous work showing that learning to generate virtual inputs via adversarial learning (GANs variants) forms better representations than simply learning to reproduce external inputs (*Radford et al., 2015*; *Donahue et al., 2016*; *Berthelot et al., 2018*).

Finally, we consider a different pathology in which REM is not driven by randomly combined episodic memories and noise, but by single episodic memories without noise, as during NREM (*Figure 4b*, purple cross). Similarly to removing REM, linear separability increases much slower across epochs, leading to worse performance of the readout (*Figure 4c and d*, purple lines). For the SVHN dataset, the performance does not reach the level of the PAD even after many wake–sleep cycles (see also *Appendix 1—figure 3d*). This suggests that combining different, possibly nonrelated episodic memories, together with spontaneous cortical activity, as reported during REM dreaming (*Fosse et al., 2003*), leads to significantly faster representation learning.

Our results suggest that generating virtual sensory inputs during REM dreaming, via a high-level combination of hippocampal memories and spontaneous cortical activity and subsequent adversarial learning, allows animals to extract semantic concepts from their sensorium. Our model provides hypotheses about the effects of REM deprivation, complementing pharmacological and optogenetic studies reporting impairments in the learning of complex rules and spatial object recognition (*Boyce et al., 2016*). For example, our model predicts that object identity would be less easily decodable from recordings of neuronal activity in the inferior-temporal (IT) cortex in animal models with chronically impaired REM sleep.

## Perturbed dreaming during NREM improves robustness of semantic representations

Generalizing beyond previously experienced stimuli is essential for an animal's survival. This generalization is required due to natural perturbations of sensory inputs, for example, partial occlusions, noise, or varying viewing angles. These alter the stimulation pattern, but in general should not change its latent representation subsequently used to make decisions.

Here, we model such sensory perturbations by silencing patches of neurons in early sensory areas during the stimulus presentation (*Figure 5a*). As before, linear separability is measured via a linear classifier that has been trained on latent representations of unoccluded images and we use stimuli that were not used during training. Adding occlusions hence directly tests the out-of-distribution generalization capabilities of the learned representations. For the model trained with all phases (*Figure 5b*, full model), the linear separability of latent representations decreases as occlusion intensity increases, until reaching chance level for fully occluded images (*Figure 5c and d*, black line).

We next consider a sleep pathology in which we suppress perturbed dreaming during the NREM phase while keeping adversarial dreaming during REM (*Figure 5b*, orange cross). Without NREM, linear separability of partially occluded images is significantly decreased for identical occlusion levels (*Figure 5c and d*; compare black and orange lines). In particular, performance degrades much faster with increasing occlusion levels. Note that despite the additional training objective the full PAD

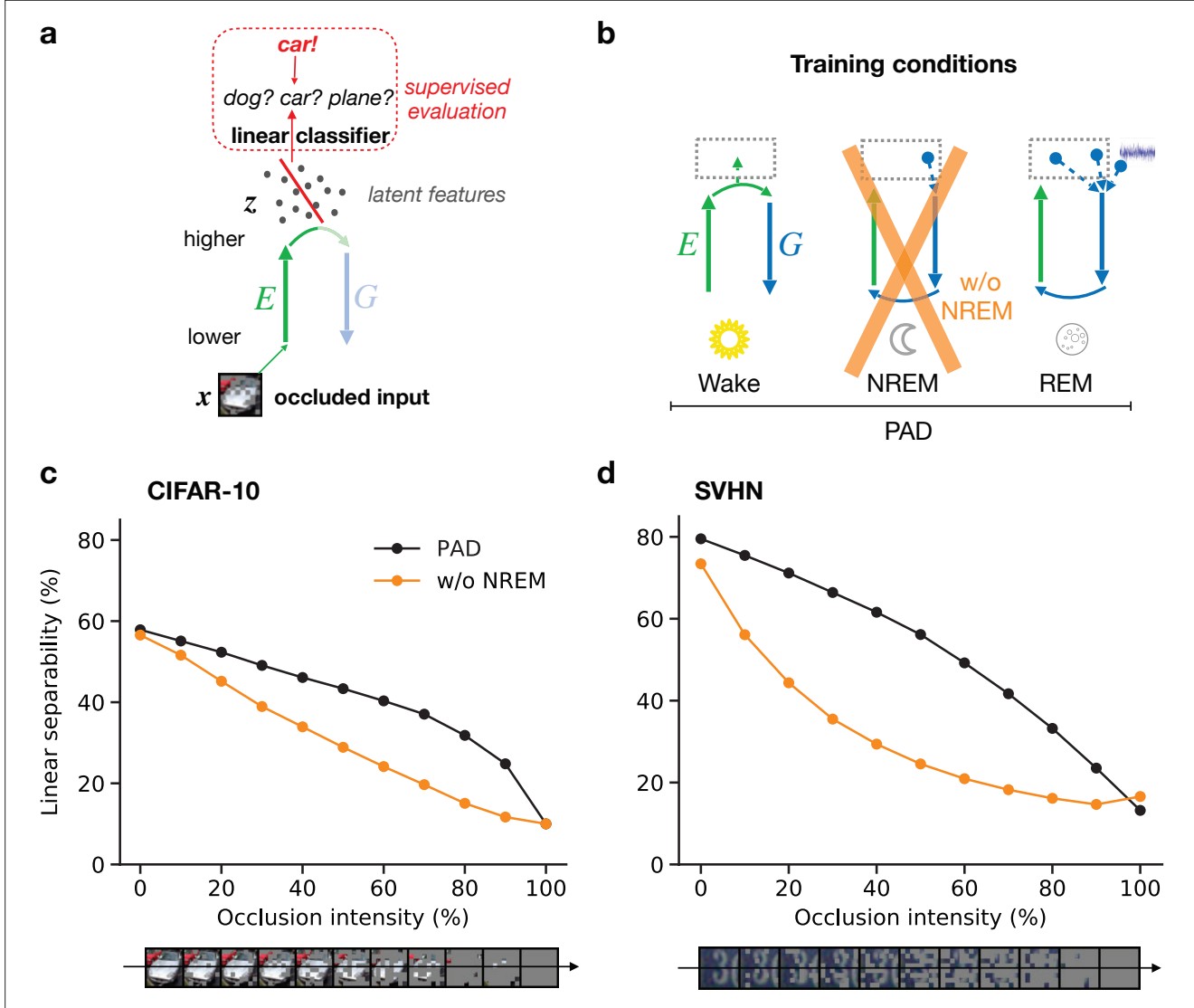

**Figure 5.** Perturbed dreaming during non-rapid eye movement (NREM) improves robustness of latent representations. (**a**) A trained linear classifier (*Figure 4*) infers class labels from latent representations. The classifier was trained on latent representations of original images, but evaluated on representations of images with varying levels of occlusion. (**b**) Training phases and pathological conditions: full model (perturbed and adversarial dreaming [PAD], black), without NREM phase (w/o NREM, orange). (**c, d**) Classification accuracy obtained on the test dataset (**c**: CIFAR-10; **d**: SVHN) after 50 epochs for different levels of occlusion (0% to 100%). Full model (PAD): black line; w/o NREM: orange line. SEM over four different initial conditions overlap with data points. Note that due to an unbalanced distribution of samples the highest performance of a naive classifier is 18.9% for the SVHN dataset.

develops equally good or even better latent representations of unoccluded images (0% occlusion intensity) compared to this pathological condition without perturbed dreams.

Crucially, the perturbed dreams in NREM are generated by replaying single episodic memories. If the latent activity fed to the generator during NREM was of similar origin as during REM, that is, obtained from a convex combination of multiple episodic memories coupled with cortical spontaneous activity, the quality of the latent representations significantly decreases (see also *Appendix 1— figure 6*). This suggests that only replaying single memories, as hypothesized to occur during NREM sleep (*O'Neill et al., 2010*), rather than their noisy combination, is beneficial to robustify latent representations against input perturbations.

This robustification originates from the training objective defined in the NREM phase, forcing feedforward pathways to map perturbed inputs to the latent representation corresponding to their clean, nonoccluded version. This procedure is reminiscent of a regularization technique from machine

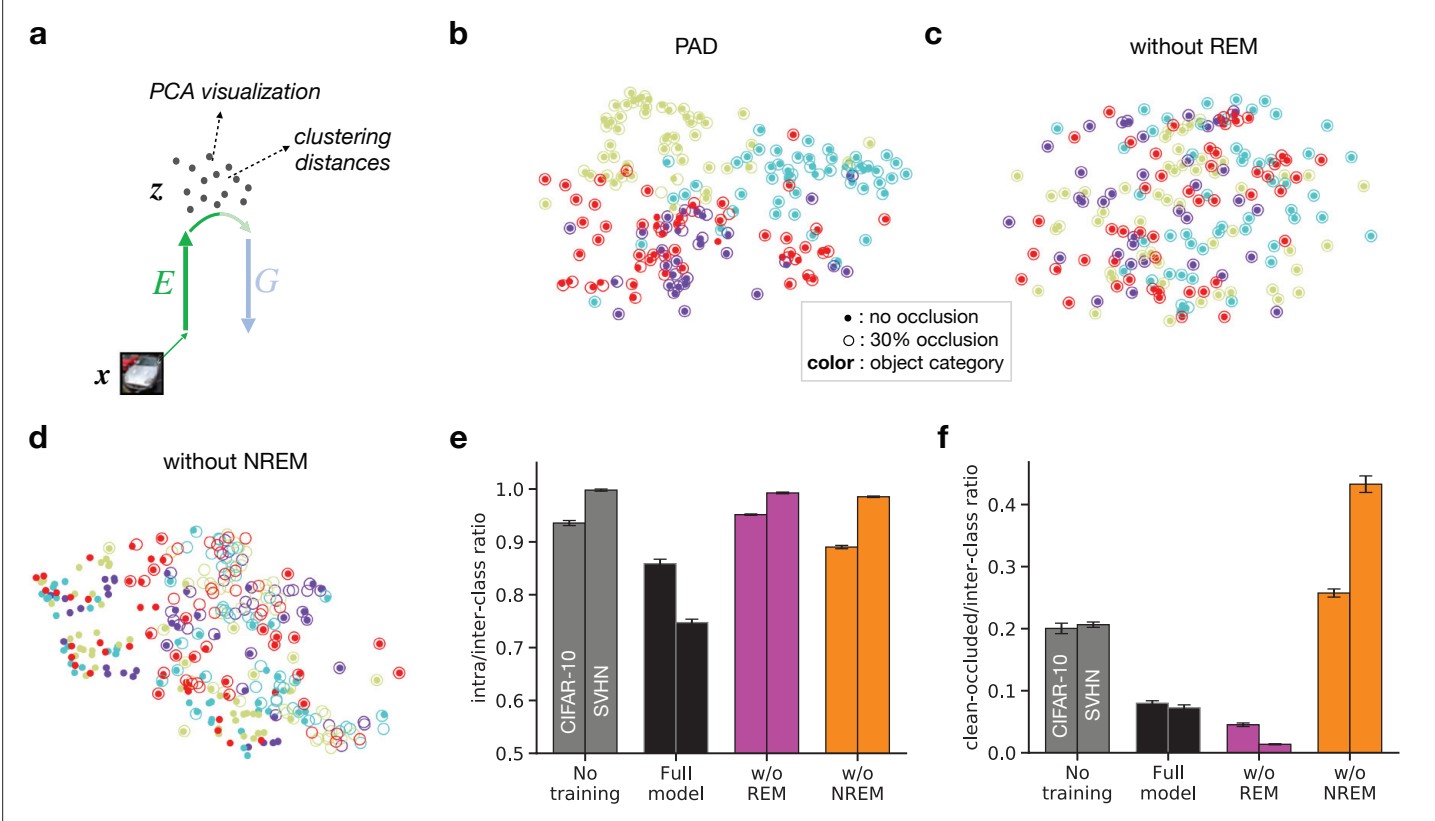

**Figure 6.** Effects of non-rapid eye movement (NREM) and rapid eye movement (REM) sleep on latent representations. (**a**) Inputs $x$ are mapped to their corresponding latent representations $z$ via the encoder $E$. Principal component analysis (PCA; *Jolliffe and Cadima, 2016*) is performed on the latent space to visualize its structure (**b–d**). Clustering distances (**e, f**) are computed directly on latent features $z$. (**b–d**) PCA visualization of latent representations projected on the first two principal components. Full circles represent clean images, open circles represent images with 30% occlusion. Each color represents an object category from the SVHN dataset (purple: '0'; cyan: '1'; yellow: '2'; red: '3'). (**e**) Ratio between average intra-class and average inter-class distances in latent space for randomly initialized networks (no training, gray), full model (black), model trained without REM sleep (w/o REM, pink), and model trained without NREM sleep (w/o NREM, orange) for unoccluded inputs. (**f**) Ratio between average clean-occluded (30% occlusion) and average inter-class distances in latent space for the full model (black), w/o REM (pink), and w/o NREM (orange). Error bars represent SEM over four different initial conditions.

learning called 'data augmentation' (*Shorten and Khoshgoftaar, 2019*), which increases the amount of training data by adding stochastic perturbations to each input sample. However, in contrast to data augmentation methods that directly operate on samples, here the system autonomously generates augmented data in offline states, preventing interference with online cognition and avoiding storage of the original samples. Our 'dream augmentation' suggests that NREM hippocampal replay not only maintains or strengthens cortical memories, as traditionally suggested (*Klinzing et al., 2019*), but also improves latent representations when only partial information is available. For example, our model predicts that animals lacking such dream augmentation, potentially due to impaired NREM sleep, fail to react reliably to partially occluded stimuli even though their responses to clean stimuli are accurate.

## Latent organization in healthy and pathological models

The results so far demonstrate that PAD, during REM and NREM sleep states, contributes to cortical representation learning by increasing the linear separability of latent representations into object classes. We next investigate how the learned latent space is organized, that is, whether representations of sensory inputs with similar semantic content are grouped together even if their low-level structure may be quite different, for example, due to different viewing angles, variations among an object category, or (partial) occlusions.

We illustrate the latent organization by projecting the latent variable $z$ using principal component analysis (PCA, *Figure 6a*, *Jolliffe and Cadima, 2016*). This method is well-suited for visualizing

| | Wake | NREM | REM |
|---|---|---|---|
| **ACh levels** | High | Low | High |
| **NA levels** | High | Low | Low |
| **Sensory activity** | Externally driven | Internally generated | Internally generated |
| **Discriminator output $d$** | High activity | Gated off | Low activity |
| **Plasticity in generator $G$** | On | Off | Sign switch |
| **Network meta-state** | Wake | Perturbed dream | Adversarial dream |
| **Phenomenology** | ☀ **Wake** | ☾ **NREM** | **REM** |

**Figure 7.** Model features and physiological counterparts during Wake, non-rapid eye movement (NREM), and rapid eye movement (REM) phases. ACh: acetylcholine; NA: noradrenaline. 'Sign switch' indicates that identical local errors lead to opposing weight changes between Wake and REM sleep.

high-dimensional data in a low-dimensional space while preserving as much of the data's variation as possible.

For PAD, the obtained PCA projection shows relatively distinct clusters of latent representations according to the semantic category ('class identity') of their corresponding images (*Figure 6b*). The model thus tends to organize latent representations such that high-level, semantic clusters are discernible. Furthermore, partially occluded objects (*Figure 6b*, empty circles) are represented close by their corresponding unoccluded version (*Figure 6b*, full circles).

As shown in the previous sections, removing either REM or NREM has a negative impact on the linear separability of sensory inputs. However, the reasons for these effects are different between REM and NREM. If REM sleep is removed from training, representations of unoccluded images are less organized according to their semantic category, but still match their corresponding occluded versions (*Figure 6c*). REM is thus necessary to organize latent representations into semantic clusters, providing an easily readable representation for downstream neurons. In contrast, removing NREM causes representations of occluded inputs to be remote from their unoccluded representations (*Figure 6d*).

We quantify these observations by computing the average distances between latent representations from the same object category (intra-class distance) and between representations of different object category (inter-class distance). Since the absolute distances are difficult to interpret, we focus on their ratio (*Figure 6e*). On both datasets, this ratio increases if the REM phase is removed from training (*Figure 6e*, compare black and pink bars), reaching levels comparable to the one with the untrained network. Moreover, removing NREM from training also increases this ratio. These observations suggest that both PAD jointly reorganize the latent space such that stimuli with a similar semantic structure are mapped to similar latent representations. In addition, we compute the distance between the latent representations inferred from clean images and their corresponding occluded

versions, also divided by the inter-class distance (*Figure 6f*). By removing NREM from training, this ratio increases significantly, highlighting the importance of NREM in making latent representations invariant to input perturbations.

## Cortical implementation of PAD

We have shown that PAD can learn semantic cortical representations useful for downstream tasks. Here, we hypothesize how the associated mechanisms may be implemented in the cortex.

First, PAD implies the existence of discriminator neurons that would learn to be differentially active during wakefulness and REM sleep. It also postulates a conductor that orchestrates learning by providing a teaching ('nudging') signal to the discriminator neurons during Wake and REM. Experimental evidence suggests that discriminator neurons, differentiating between internally generated end externally driven sensory activity, may reside in the anterior cingulate cortex or the medial prefrontal cortex, but functionally similar neurons may be located across cortex to deliver local learning signals (*Subramaniam et al., 2012*; *Simons et al., 2017*; *Gershman, 2019*; *Benjamin and Kording, 2021*).

Second, learning in PAD is orchestrated across three different phases: (1) learning stimulus reconstruction during Wake, (2) learning latent variable reconstruction during NREM sleep ('perturbed dreaming'), and (3) learning to generate realistic sensory activity during REM sleep ('adversarial dreaming'). Our model suggests that objective functions and synaptic plasticity are affected by these phases (*Figure 7*). Wakefulness is associated with increased activity of modulatory brainstem neurons releasing neuromodulators such as acetylcholine (ACh) and noradrenaline (NA), hypothesized to prioritize the amplification of information from external stimuli (*Adamantidis et al., 2019*; *Aru et al., 2020*). In contrast, neuromodulator concentrations during NREM are reduced compared to Wake, while REM is characterized by high ACh and low NA levels (*Hobson, 2009*). We postulate that the state-specific modulation provides a high activity target for the discriminator during Wake, which is decreased during REM and entirely gated off during NREM. Furthermore, we suggest that adversarial learning is implemented by a sign-switched plasticity in the generative network during REM sleep, with respect to Wake. During wakefulness, plasticity in these apical synapses may be enhanced by NA as opposed to NREM (*Adamantidis et al., 2019*; *Aru et al., 2020*). The presence of ACh alone during REM (*Hobson et al., 2000*) may switch the sign of plasticity in apical synapses of (hippocampal) pyramidal neurons (*McKay et al., 2007*). Furthermore, it is known that somato-dendritic synchrony is reduced in REM versus NREM sleep (*Seibt et al., 2017*); this suggests a reduced somato-dendritic backpropagation of action potentials, which, in turn, is known to switch the sign of apical plasticity (*Sjöström and Häusser, 2006*).

Third, learning in our model requires the computation of reconstruction errors, that is, mismatches between top-down and bottom-up activity. So far, two nonexclusive candidates for computing mismatch signals have been proposed. One suggests a dendritic error representation in layer 5 pyramidal neurons that compare bottom-up with top-down inputs from our encoding ($E$) and generative ($G$) pathways (*Guerguiev et al., 2017*; *Sacramento et al., 2018*). The other suggests an explicit mismatch representation by subclasses of layer 2/3 pyramidal neurons (*Keller and Mrsic-Flogel, 2018*).

Fourth, our computational framework assumes effectively separate feedforward and feedback streams. A functional separation of these streams does not necessarily imply a structural separation at the network level. Indeed, such cross-projections are observed in experimental data (*Gilbert and Li, 2013*) and also used in, for example, the predictive processing framework (*Rao and Ballard, 1999*). In our model, an effective separation of the information flows is required to prevent 'information shortcuts' across early sensory cortices that would prevent learning of good representations in higher sensory areas. This suggests that for significant periods of time intra-areal lateral interactions between cortical feedforward and feedback pathways are effectively gated off in most of the areas.

Fifth, similar to previous work (*Káli and Dayan, 2004*), the hippocampus is not explicitly modeled but rather mimicked by a buffer allowing simple store and retrieve operations. An extension of our model could replace this simple mechanism with attractor networks that have been previously employed to model hippocampal function (*Tang et al., 2010*). The combination of episodic memories underlying REM dreams in our model could either occur in the hippocampus or in the cortex. In either case, we would predict a nearly simultaneous activation of different episodic memories in the hippocampus that results in the generation of creative virtual early cortical activity.

Finally, beyond the mechanisms discussed above, our model assumes that cortical circuits can efficiently perform credit assignment, similar to the classical error backpropagation algorithm. Most biologically plausible implementations for error backpropagation involve feedback connections to deliver error signals (*Whittington and Bogacz, 2019*; *Richards et al., 2019*; *Lillicrap et al., 2020*), for example, to the apical dendrites of pyramidal neurons (*Sacramento et al., 2018*; *Guerguiev et al., 2017*; *Haider et al., 2021*). An implementation of our model in such a framework would hence require additional feedforward and feedback connections for each neuron. For example, neurons in the feedforward pathway would not only project to higher cortical areas to transmit signals, but additionally project back to earlier areas to allow these to compute the local errors required for effective learning. Overall, our proposed model could be mechanistically implemented in cortical networks through different classes of pyramidal neurons with a biological version of supervised learning based on a dendritic prediction of somatic activity (*Urbanczik and Senn, 2014*), and a corresponding global modulation of synaptic plasticity by state-specific neuromodulators.

## Discussion

Semantic representations in cortical networks emerge in early life despite most observations lacking an explicit class label, and sleep has been hypothesized to facilitate this process (*Klinzing et al., 2019*). However, the role of dreams in cortical representation learning remains unclear. Here, we proposed that creating virtual sensory experiences by randomly combining episodic memories during REM sleep lies at the heart of cortical representation learning. Based on a functional cortical architecture, we introduced the PAD model and demonstrated that REM sleep can implement an adversarial learning process that, constrained by the network architecture and the choice of latent prior distributions, builds semantically organized latent representations. Additionally, perturbed dreaming based on the episodic memory replay during NREM stabilizes the cortical representations against sensory perturbations. Our computational framework allowed us to investigate the effects of specific sleep-related pathologies on cortical representations. Together, our results demonstrate complementary effects of perturbed dreaming from individual episodes during NREM and adversarial dreaming from mixed episodes during REM. PAD suggests that the generalization abilities exhibited by humans and other animals arise from distinct processes during the two sleep phases: REM dreams organize representations semantically and NREM dreams stabilize these representations against perturbations. Finally, the model suggests how adversarial learning inspired by GANs can potentially be implemented by cortical circuits and associated plasticity mechanisms.

### Relation to cognitive theories of sleep

PAD focuses on the functional role of sleep, and in particular dreams. Many dynamic features of brain states during NREM and REM sleep, such as cortical oscillations (*Léger et al., 2018*), are hence ignored here but will potentially become relevant when constructing detailed circuit models of the suggested architectures, for example, for switching between memories (*Korcsak-Gorzo et al., 2021*). Our proposed model of sleep is complementary to theories suggesting that sleep is important for physiological and cognitive maintenance (*McClelland et al., 1995*; *Káli and Dayan, 2004*; *Rennó-Costa et al., 2019*; *van de Ven et al., 2020*). In particular, *Norman et al., 2005* proposed a model where autonomous reactivation of memories (from cortex and hippocampus) coupled with oscillating inhibition during REM sleep helps detect weak parts of memories and selectively strengthen them to overcome catastrophic forgetting. While our REM phase serves different purposes, an interesting commonality is the view of REM as a period where the cortex 'thinks about what it already knows' from past and recent memories and reorganizes its representations by replaying them together, as opposed to NREM where only recent memories are replayed and consolidated. Recent work has also suggested that the brain learns using adversarial principles either as a reality monitoring mechanism potentially explaining delusions in some mental disorders (*Gershman, 2019*), in the context of dreams to overcome overfitting and promote generalization (*Hoel, 2021*), or for learning inference in recurrent biological networks (*Benjamin and Kording, 2021*).

Cognitive theories propose that sleep promotes the abstraction of semantic concepts from episodic memories through a hippocampo-cortical replay of waking experiences, referred to as 'memory semantization' (*Nadel and Moscovitch, 1997*; *Lewis and Durrant, 2011*). The learning of

organized representations is an important basis for semantization. An extension of our model would consider the influence of different sensory modalities on representation learning (*Guo et al., 2019*), which is known to significantly influence cortical schemas (*Lewis et al., 2018*) and can encourage the formation of computationally powerful representations (*Radford et al., 2021*).

Finally, sleep has previously been considered as a state where 'noisy' connections acquired during wakefulness are selectively forgotten (*Crick and Mitchison, 1983*; *Poe, 2017*), or similarly, as a homeostatic process to desaturate learning and renormalize synaptic strength (synaptic homeostasis hypothesis; *Tononi and Cirelli, 2014*; *Tononi and Cirelli, 2020*). In contrast, our model offers an additional interpretation of plasticity during sleep, where synapses are globally readapted to satisfy different but complementary learning objectives than Wake, either by improving feedforward recognition of perturbed inputs (NREM) or by adversarially tuning top-down generation (REM).

## Relation to representation learning models

Recent advances in machine learning, such as self-supervised learning approaches, have provided powerful techniques to extract semantic information from complex datasets (*Liu et al., 2021*). Here, we mainly took inspiration from self-supervised generative models combining autoencoder and adversarial learning approaches (*Radford et al., 2015*; *Donahue et al., 2016*; *Dumoulin et al., 2017*; *Berthelot et al., 2018*; *Liu et al., 2021*). It is theoretically not yet fully understood how linearly separable representations are learned from objectives that do not explicitly encourage them, that is, reconstruction and adversarial losses. We hypothesize that the presence of architectural constraints and latent priors, in combination with our objectives, enables their emergence (see also *Alemi et al., 2018*; *Tschannen et al., 2020*). Note that similar generative machine learning models often report a higher linear separability of network representations, but use all convolutional layers as a basis for the readout (*Radford et al., 2015*; *Dumoulin et al., 2017*), while we only used low-dimensional features $z$. Approaches similar to ours, that is, those that perform classification only on the latent features, report comparable performance to ours (*Berthelot et al., 2018*; *Hjelm et al., 2019*; *Beckham et al., 2019*).

Furthermore, in contrast to previous GAN variants, our model removes many optimization tricks such as batch normalization layers (*Ioffe and Szegedy, 2015*), spectral normalization layers (*Miyato et al., 2018*), or optimizing the min-max GAN objective in three steps with different objectives, which are challenging to implement in biological substrates. Despite their absence, our model maintains a high quality of latent representations. As our model is relatively simple, it is amenable to implementations within frameworks approximating backpropagation in the brain (*Whittington and Bogacz, 2019*; *Richards et al., 2019*; *Lillicrap et al., 2020*). However, some components remain challenging for implementations in biological substrates, for example, convolutional layers (but see *Pogodin et al., 2021*) and batched training (but see *Marblestone et al., 2016*).

## Dream augmentations, mixing strategies, and fine-tuning

To make representations robust, a computational strategy consists of learning to map different sensory inputs containing the same object to the same latent representation, a procedure reminiscent of data augmentation (*Shorten and Khoshgoftaar, 2019*). As mentioned above, unlike standard data augmentation methods, our NREM phase does not require the storage of raw sensory inputs to create altered inputs necessary for such data augmentation and instead relies on (hippocampal) replay being able to regenerate similar inputs from high-level representations stored during wakefulness. Our results obtained through perturbed dreaming during NREM provide initial evidence that this dream augmentation may robustify cortical representations.

Furthermore, as discussed above, introducing more specific modifications of the replayed activity, for example, mimicking translations or rotations of objects, coupled with a negative phase where latent representations from different images are pushed apart, may further contribute to the formation of invariant representations. Along this line, recent self-supervised contrastive learning methods (*Gidaris et al., 2018*; *Chen et al., 2020*; *Zbontar et al., 2021*) have been shown to enhance the semantic structure of latent representations by using a similarity objective where representations of stimuli under different views are pulled together in a first phase, while, crucially, embedding distances between unrelated images are increased in a second phase.

In our REM phase, different mixing strategies in the latent layer could be considered. For instance, latent activities could be mixed up by retaining some vector components of a representation and using the rest from a second one (*Beckham et al., 2019*). Moreover, more than two memory representations could have been used. Alternatively, our model could be trained with spontaneous cortical activity only. In our experimental setting, we do not observe significant differences between using a combination of episodic memories with spontaneous activity or only using spontaneous activity (*Appendix 1—figure 4*). However, we hypothesize that for models that learn continuously, a preferential replay of combinations of recent episodic memories encourages the formation of cortical representations that are useful in the present.

Here, we used a simple linear classifier to measure the quality of latent representations, which is an obvious simplification with regard to cortical processing. Note, however, that also for more complex 'readouts' organized latent representations enable more efficient and faster learning (*Silver et al., 2017*; *Ha and Schmidhuber, 2018*; *Schrittwieser et al., 2020*). In its current form, PAD assumes that training the linear readout does not lead to weight changes in the encoder network. However, in cortical networks, cognitive or motor tasks leveraging latent representations likely shape the encoder network, which could in our model be reflected in 'fine-tuning' the encoder for specific tasks (compare *Liu et al., 2021*).

Finally, our model does not show significant differences in performance when the order of sleep phases is switched (*Appendix 1—figure 5*). However, NREM and REM are observed to occur in a specific order throughout the night (*Diekelmann and Born, 2010*), and this order has been hypothesized to be important for memory consolidation ('sequential hypothesis,' *Giuditta et al., 1995*). The independence of phases in our model may be due to the relatively small synaptic changes occurring in each phase. We expect the order of sleep phases to influence model performance if these changes become larger either due to longer phases or increased learning rates. The latter may become particularly relevant in continual learning settings where it becomes important to control the emphasis put on recent observations.

## Signatures of generative learning

PAD makes several experimentally testable predictions at the neuronal and systems level. We first address generally whether the brain learns via generative models during sleep before discussing specific signatures of adversarial learning.

First, our NREM phase assumes that hippocampal replay generates perturbed wake-like early sensory activity (see also *Ji and Wilson, 2007*), which is subsequently processed by feedforward pathways. Moreover, our model predicts that over the course of learning sensory-evoked neuronal activity and internally generated activity during sleep become more similar. In particular, we predict that (spatial) activity in both NREM and REM become more similar to Wake; however, patterns observed during REM remain distinctly different due to the creative combination of episodic memories. Future experimental studies could confirm these hypotheses by recording early sensory activity during wakefulness, NREM, and REM sleep at different developmental stages and evaluating commonalities and differences between activity patterns. Previous work has already demonstrated increasing similarity between stimulus-evoked and spontaneous (generated) activity patterns during wakefulness in the ferret visual cortex (*Berkes et al., 2011*; but see *Avitan et al., 2021*).

On a behavioral level, the improvement of internally generated activity patterns correlates with the development of dreams in children, which are initially unstructured, simple, and plain, and gradually become full-fledged, meaningful, narrative, implicating known characters and reflecting life episodes (*Nir and Tononi, 2010*). In spite of their increase in realism, REM dreams in adulthood are still reported as bizarre (*Williams et al., 1992*). Bizarre dreams, such as 'flying dogs,' are typically defined as discontinuities or incongruities of the sensory experience (*Mamelak and Hobson, 1989*) rather than completely structureless experiences. This definition hence focuses on high-level logical structure, not on the low-level sensory content. In contrast, the low FID score, that is, high realism, of REM dreams in our experiments reflects that the low-level structure on which this evaluation metric mainly focuses (e.g., *Brendel and Bethge, 2019*) is similar to actual sensory input. Capturing the 'logical realism' of our generated neuronal activities most likely requires a more sophisticated evaluation metric and an extension of the model capable of generating temporal sequences of sensory stimulation. We note, however, that even such surreal dreams as 'flying dogs'

can be interpreted as altered combinations of episodic memories and thus, in principle, can arise from our model.

Second, our model suggests that the development of semantic representations is mainly driven by REM sleep. This allows us to make predictions that connect the network with the systems level, in the specific case of acquiring skills from complex and unfamiliar sensory input. For humans, this could be learning a foreign language with unfamiliar phonetics. Initially, cortical representations cannot reflect relevant nuances in these sounds. Phonetic representations develop gradually over experience and are reflected in changes of the sensory-evoked latent activity, specifically in the reallocation of neuronal resources to represent the relevant latent dimensions. We hypothesize that in the case of impaired REM sleep this change of latent representations is significantly reduced, which goes hand in hand with decreased learning speed. Future experimental studies could investigate these effects, for instance, by trying to decode sound identity from high-level cortical areas in patients where REM sleep is impaired over long periods through pharmacological agents such as antidepressants (**Boyce et al., 2017**). An equivalent task in the nonhuman animal domain would be song acquisition in songbirds (**Fiete et al., 2007**). On a neuronal level, one could selectively silence feedback pathways during REM sleep in animal models over many nights, for example, via optogenetic tools. Our model predicts that this silencing would significantly impact the animal's learning speed, as reported from animals with reduced theta rhythm during REM sleep (**Boyce et al., 2017**).

## Signatures of adversarial learning

The experimental predictions discussed above mainly address whether the brain learns via generative models during sleep. Here, we make experimental predictions that would support our hypotheses and contrast them to alternative theories of learning during offline states.

### Existence of an external/internal discriminator

The discriminator provides our model with the ability to distinguish externally driven from internally generated low-level cortical activity. Due to this unique property, the discriminator may be leveraged to distinguish actual from imagined sensations. According to our model, reduced REM sleep would lead to an impaired discriminator, and could thus result in an inability of subjects to realize that self-generated imagery is not part of the external sensorium. This may result in the formation of delusions, as previously suggested (**Gershman, 2019**). For instance, hallucinations in schizophrenic patients, often mistaken for veridical perceptions (**Waters et al., 2016**), could be partially caused by abnormal REM sleep patterns, related to observed reduced REM latency and density (**Cohrs, 2008**). Based on these observations, we predict in the context of our model a negative correlation between REM sleep quality and delusional perceptions of hallucinations. Systematic differences in REM sleep quality may hence explain why some patients are able to recognize that their hallucinations are self-generated while some others mistake them to be real. Moreover, although locating discriminator neurons may prove nontrivial (but see 'Cortical implementation of PAD' for specific suggestions), we predict that once the relevant cells have been identified, perturbing them may lead to detrimental effects on differentiating between external sensory inputs and internally generated percepts.

The state-specific activity of the discriminator population makes predictions about plasticity on synapses in the feedforward stream during wakefulness and sleep. In our model, the discriminator is trained to distinguish externally from internally generated patterns by opposed targets imposed during Wake and REM. After many wake–sleep cycles, the KL loss as well as the reconstruction loss (see Materials and methods) in our model become small compared to the adversarial loss (**Appendix 1—figure 1**, **Appendix 1—figure 2**), which remains nonzero due to a balance between discriminator and generator. The same low-level activity pattern would hence cause opposite weight changes during wakefulness and sleep on feedforward synapses. This could be tested experimentally by actively instantiating similar spatial activity patterns in low-level sensory cortex during wakefulness and REM and comparing the statistics of observed changes in (feedforward) downstream synapses.

### Adversarial training of a generator during sleep

To drive adversarial learning and maintain a balance between the generator and discriminator, the generative network must be trained in parallel to the discriminative (encoder) network during REM. In contrast, in alternative representation learning models that involve offline states such as the

Wake–Sleep algorithm (**Hinton et al., 1995**), generative pathways are not trained to produce realistic dreams during the sleep phase. Rather, they are trained by reconstruction on real input data during the wake phase. This allows an experimental distinction between our model and Wake–Sleep-like models: while our model predicts plasticity in both bottom-up and top-down pathways both during wake and during REM sleep, Wake–Sleep models alternate between training feedback and feedforward connections during online and offline states, respectively.

Previous work has developed methods to infer plasticity rules from neuronal activity (**Lim et al., 2015**; **Senn and Sacramento, 2015**) or weight changes (**Nayebi et al., 2020**). In the spirit of existing in vivo experiments, we suggest to optogenetically monitor and potentially modulate apical dendritic activities in cortical pyramidal neurons of mice during wakefulness and REM sleep (**Li et al., 2017**; **Voigts and Harnett, 2020**; **Schoenfeld et al., 2022**). From the statistics of the recorded dendritic and neuronal activity, the plasticity rules could be inferred and compared to the state-dependent rules suggested by our model, particularly to the predicted sign switch of plasticity between wakefulness and REM sleep.

## Adversarial learning and creativity

Adversarial learning, for example, in GANs, enables a form of creativity, reflected in their ability to generate realistic new data or to create semantically meaningful interpolations (**Radford et al., 2015**; **Berthelot et al., 2018**; **Karras et al., 2018**). This creativity might be partly caused by the freedom in generating sensory activity that is not restricted by requiring good reconstructions, but is only guided by the internal/external judgment (**Goodfellow, 2016**). This is less constraining on the generator than direct reconstruction losses used in alternative models such as variational auto-encoders (**Kingma and Welling, 2013**) or the Wake–Sleep algorithm (**Hinton et al., 1995**). We thus predict that REM sleep, here implementing adversarial learning, should boost creativity, as previously reported (**Cai et al., 2009**; **Llewellyn, 2016**; **Lewis et al., 2018**). Furthermore, we predict that REM sleep influences a subject's ability to visualize creative mental images, for instance, associating nonobvious visual patterns with distinct memories. For example, we predict that participants chronically lacking REM sleep would perform worse than control participants at a creative synthesis task (**Palmiero et al., 2015**), consisting of combining different visual components into a new, potentially useful object.

## Adversarial learning and lucid dreaming

Finally, adversarial dreaming offers a theoretical framework to investigate neuronal correlates of normal versus lucid dreaming (**Dresler et al., 2012**; **Baird et al., 2019**). While in normal dreaming the internally generated activity is perceived as externally caused, in lucid dreaming it is perceived as what it is, that is, internally generated. We hypothesize that the 'neuronal conductor' that orchestrates adversarial dreaming is also involved in lucid dreaming by providing to the dreamer conscious access to the target 'internal' that the conductor imposes during REM sleep. Our cortical implementation suggests that the neuronal conductor could gate the discriminator teaching via the apical activity of cortical pyramidal neurons. The same apical dendrites were also speculated to be involved in conscious perception (**Takahashi et al., 2020**), dreaming (**Aru et al., 2020**), and in representing the state and content of consciousness (**Aru et al., 2019**).

Our model demonstrates that adversarial learning during wakefulness and sleep can provide significant benefits to extract semantic concepts from sensory experience. By bringing insights from modern artificial intelligence to cognitive theories of sleep function, we suggest that cortical representation learning during dreaming is a creative process, orchestrated by brain-state-regulated adversarial games between separated feedforward and feedback streams. Adversarial dreaming may further be helpful to understand learning beyond the standard student–teacher paradigm. By 'seeing' the world from new perspectives every night, dreaming represents an active learning phenomenon, constantly improving our understanding, our creativity, and our awareness.

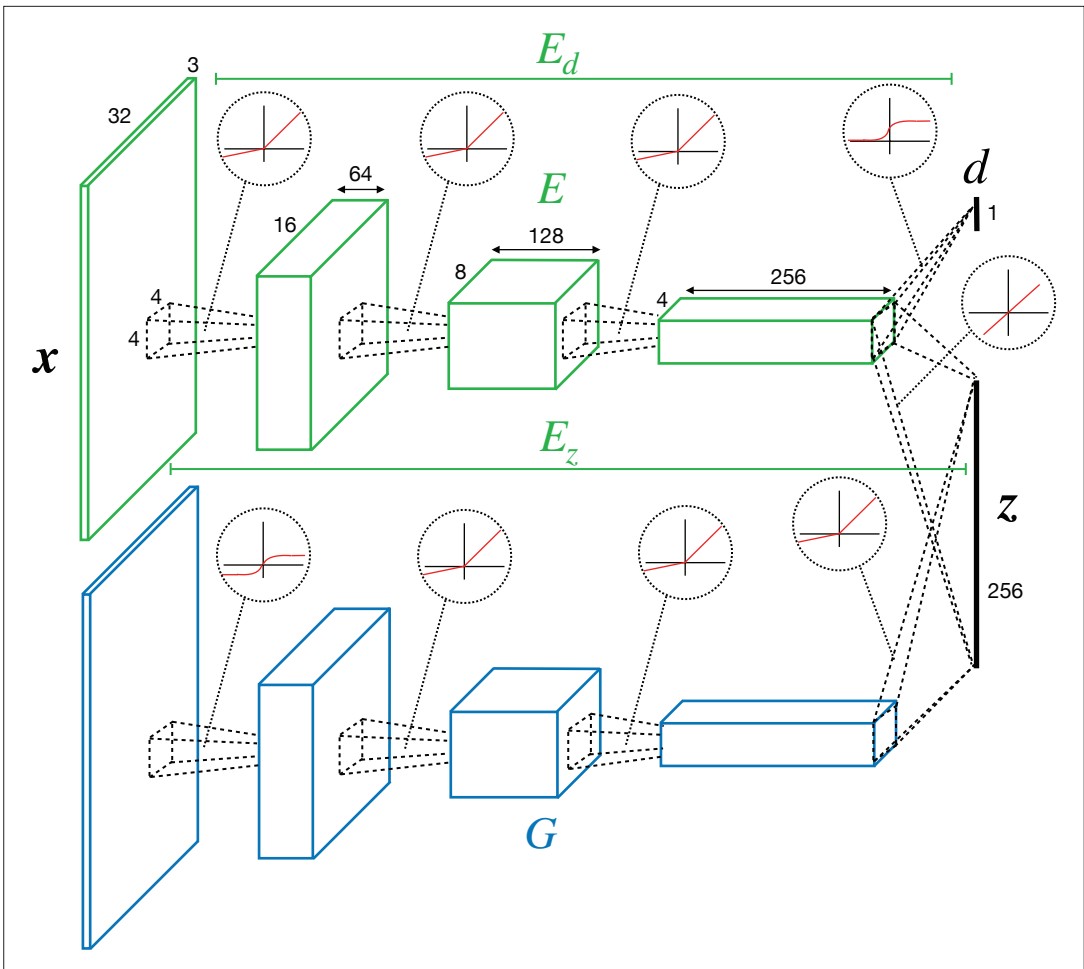

**Figure 8.** Convolutional neural network (CNN) architecture of encoder/discriminator and generator used in perturbed and adversarial dreaming (PAD).

## Materials and methods
### Network architecture

The network consists of two separate pathways, mapping from the pixel to the latent space ('encoder'/'discriminator') and from the latent to pixel space ('generator'). Encoder/discriminator and generator architectures follow a similar structure as the Deep Convolutional Generative Adversarial Networks (DCGANs) model (*Radford et al., 2015*). The encoder $E_z$ has four convolutional layers (*LeCun et al., 2015*) containing 64, 128, 256, and 256 channels, respectively (*Figure 8*).

Each layer uses a 4 × 4 kernel, a padding of 1 (0 for last layer), and a stride of 2, that is, feature size is halved in each layer. All convolutional layers except the last one are followed by a LeakyReLU nonlinearity (*Maas et al., 2013*). We denote the activity in the last convolutional layer as $z$. An additional convolutional layer followed by a sigmoid nonlinearity is added on top of the second-to-last layer of the encoder and maps to a single scalar value $d$, the internal/external discrimination (with putative teaching signal 0 or 1). We denote the mapping from $x$ to $d$ by $E_d$. $E_z$ and $E_d$ thus share the first three convolutional layers. We jointly denote them by $E$, where $E(x) = (E_z(x), E_d(x)) = (z, d)$ (*Figure 8*).

Mirroring the structure of $E_z$, the generator $G$ has four deconvolutional layers containing 256, 128, 64, and 3 channels. They all use a 4 × 4 kernel, a padding of 1 (0 for first deconvolutional layer), and a stride of 2, that is, the feature size is doubled in each layer. The first three deconvolutional layers are followed by a LeakyReLU nonlinearity, and the last one by a tanh nonlinearity.

As a detailed hippocampus model is outside the scope of this study, we mimic hippocampal storage and retrieval by storing and reading latent representations to and from memory.

## Datasets

We use the CIFAR-10 (*Krizhevsky and Hinton, 2009*) and SVHN (*Netzer et al., 2011*) datasets to evaluate our model. They consist of 32 × 32 pixel images with three color channels. We consider their usual split into a training set and a smaller test set.

## Training procedure

We train our model by performing stochastic gradient descent with mini-batches on condition-specific objective functions, in the following also referred to as loss functions, using the ADAM optimizer ($\beta_1 = 0.5$, $\beta_2 = 0.999$; *Kingma and Ba, 2017*) with a learning rate of 0.0002 and mini-batch size of 64. We rely on our model being fully differentiable. The following section describes the loss functions for the respective conditions.

**Algorithm 1: Training procedure**

$\theta_E, \theta_G$; //initialize network parameters

**for** *number of training iterations* **do**

 Wake
 $X \leftarrow \{x^{(1)}, ..., x^{(b)}\}$ //random mini-batch from dataset
 $Z, D \leftarrow E(X)$ //infer latent and discriminative outputs
 $X' \leftarrow G(Z)$ //reconstruct input via generator

 $\mathcal{L}_{\text{img}} \leftarrow \frac{1}{b}\sum_{i=1}^{b} \|x^{(i)} - x'^{(i)}\|^2$ //compute reconstruction loss

 $\mathcal{L}_{\text{KL}} \leftarrow \text{D}_{\text{KL}}(q(Z)\|p(Z))$ //compute KL-loss

 $\mathcal{L}_{\text{real}} \leftarrow -\frac{1}{b}\sum_{i=1}^{b} \log(d^{(i)})$ //compute discriminator loss on real samples
 $\theta_E \leftarrow \theta_E - \nabla_{\theta_E}(\mathcal{L}_{\text{img}} + \mathcal{L}_{\text{KL}} + \mathcal{L}_{\text{real}})$ //update encoder/discriminator parameters
 $\theta_G \leftarrow \theta_G - \nabla_{\theta_G}\mathcal{L}_{\text{img}}$ //update generator parameters

 NREM sleep
 $Z \leftarrow \{z^{(1)}, ..., z^{(b)}\}$ //mini-batch of latent vectors from Wake
 $X' \leftarrow G(Z)$ //reconstruct input via generator
 $Z' \leftarrow E_z(X' \odot \Omega)$ //infer perturbed input>

 $\mathcal{L}_{\text{NREM}} \leftarrow \frac{1}{b}\sum_{i=1}^{b} \|z^{(i)} - z'^{(i)}\|^2$ //compute reconstruction loss
 $\theta_E \leftarrow \theta_E - \nabla_{\theta_E}\mathcal{L}_{\text{NREM}}$

 REM sleep
 **if** *first iteration* **then**
 $Z_{\text{mix}} \leftarrow Z$
 **else**
 $Z_{\text{mix}} \leftarrow \lambda'(\lambda Z + (1-\lambda)Z_{\text{old}}) + (1-\lambda')\epsilon$ //convex combination of current and old latent vectors with noise
 **end**
 $D \leftarrow E_d(G(Z_{\text{mix}}))$

 $\mathcal{L}_{\text{REM}} \leftarrow -\frac{1}{b}\sum_{i=1}^{b} \log\left(1 - d^{(i)}\right)$ //compute adversarial loss

 $\theta_E \leftarrow \theta_E - \nabla_{\theta_E}\mathcal{L}_{\text{REM}}$
 $\theta_G \leftarrow \theta_G + \nabla_{\theta_G}\mathcal{L}_{\text{REM}}$ //gradient ascent on discriminator loss
 $Z_{\text{old}} \leftarrow Z$ //keep current vectors for next iteration

**end**

## Loss functions

### Wake

In the Wake condition, we minimize the following objective function, composed of a loss for image encoding, a regularization, and a real/fake (external/internal) discriminator,

$$\mathcal{L}_{\text{Wake}} = \mathcal{L}_{\text{img}} + \mathcal{L}_{\text{KL}} + \mathcal{L}_{\text{real}} . \tag{1}$$

$E_z$ and $G$ learn to reconstruct the mini-batch of images $X = \{x^{(1)}, x^{(2)}, ..., x^{(b)}\}$ similarly to autoencoders (**Bengio et al., 2013**) by minimizing the image reconstruction loss $\mathcal{L}_{\text{img}}$ defined by

$$\mathcal{L}_{\text{img}} = \frac{1}{b} \sum_{i=1}^{b} \| x^{(i)} - G(E_z(x^{(i)})) \|^2 \, ,$$

(2)

where $b$ denotes the size of the mini-batch. We store the latent vectors $Z = E_z(X)$ corresponding to the current mini-batch for usage during the NREM and REM phases.

We additionally impose a Kullback–Leibler divergence loss on the encoder $E_z$. This acts as a regularizer and encourages latent activities to be Gaussian with zero mean and unit variance:

$$\mathcal{L}_{\text{KL}} = D_{\text{KL}}(q(Z \,|\, X) \,\|\, p(Z)) \, ,$$

(3)

where $q(Z|X) \sim \mathcal{N}(\mu, \sigma^2)$ is a distribution over the latent variables $Z$, parameterized by mean μ and standard deviation $\sigma$, and $p(Z) \sim \mathcal{N}(0, 1)$ is the prior distribution over latent variables. $E_z$ is trained to minimize the following loss:

$$\mathcal{L}_{\text{KL}} = \frac{1}{2n_z} \sum_{j=1}^{n_z} \left( \mu_j^{(Z)2} + \sigma_j^{(Z)2} - 1 - \log(\sigma_j^{(Z)2}) \right) \, ,$$

(4)

where $n_z$ denotes the dimension of the latent space and where $\mu_j^{(Z)}$ and $\sigma_j^{(Z)}$ represent the $j^{\text{th}}$ elements of respectively the empirical mean $\mu^{(Z)}$ and empirical standard deviation $\sigma^{(Z)}$ of the set of latent vectors $E_z(X) = Z$.

As part of the adversarial game, $E_d$ is trained to classify the mini-batch of images as real. This corresponds to minimizing the loss defined as sum across the mini-batch size $b$,

$$\mathcal{L}_{\text{real}} = \mathcal{L}_{\text{GAN}}(E_d(X), 1) = -\frac{1}{b} \sum_{i=1}^{b} \log(E_d(x^{(i)})) \, .$$

(5)

Note that, in principle, $\mathcal{L}_{\text{GAN}}$ can be any GAN-specific loss function (**Gui et al., 2020**). Here, we choose the binary cross-entropy loss.

## NREM sleep

Each Wake phase is followed by an NREM phase. During this phase, we make use of the mini-batch of latent vectors $z$ stored during the Wake phase. Starting from a mini-batch of latent vectors, we generate images $G(z)$. Each obtained image of $G(z)$ is multiplied by a binary occlusion mask $\omega$ of the same dimension. This mask is generated by randomly picking two occlusion parameters, occlusion intensity and square size (for details, see section 'Image occlusion'). The encoder $E_z$ learns to reconstruct the latent vectors $z$ by minimizing the following reconstruction loss:

$$\mathcal{L}_{\text{NREM}} = \frac{1}{b} \sum_{i=1}^{b} \| z^{(i)} - E_z \left( G(z^{(i)}) \odot \omega \right) \|^2 \, ,$$

(6)

where $\odot$ denotes the element-wise product.

## REM sleep

In REM, each latent vector from the mini-batch considered during Wake is combined with the latent vector from the previous mini-batch, the whole being convex combined with a mini-batch of noise vectors $\epsilon \sim \mathcal{N}(0, I)$, where $I$ is the identity matrix, leading to a mini-batch of latent vectors $Z_{\text{mix}} = \lambda'(\lambda Z + (1 - \lambda)Z_{\text{old}}) + (1 - \lambda')\epsilon$. Here, $\lambda = 0.5$ and $\lambda' = 0.5$, where $Z_{\text{old}}$ is the previous mini-batch of latent activities. This batch of latent vectors is passed through $G$ to generate the associated images $G(Z_{\text{mix}})$. In this phase, the loss function encourages $E_d$ to classify $G(Z_{\text{mix}})$ as fake, while adversarially pushing $G$ to generate images that are less likely to be classified as fake by the minimax objective

$$\min_{E_d} \max_{G} \mathcal{L}_{\text{REM}} \, ,$$

(7)

where

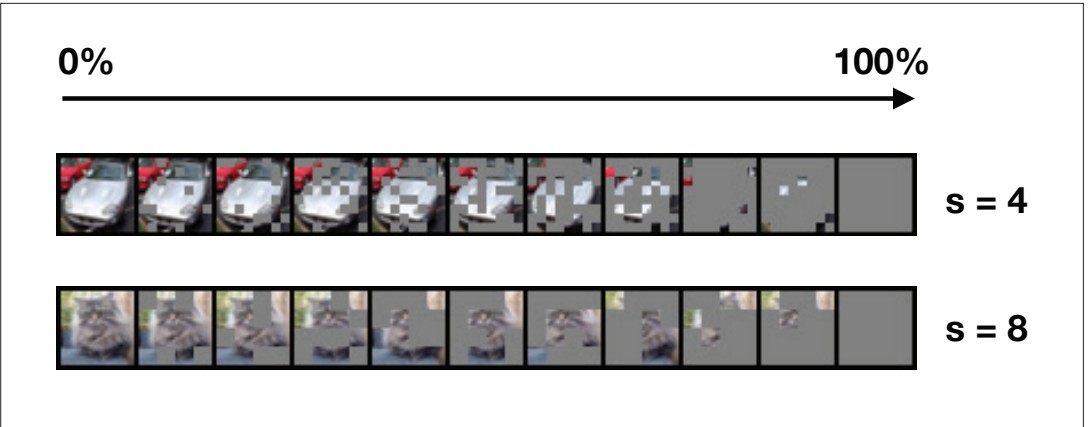

**Figure 9.** Varying size and intensity of occlusions on example images from CIFAR-10. Image occlusions vary along two parameters: occlusion intensity, defined by the probability to apply a gray square at a given position, and square size (s).

$$\mathcal{L}_{\text{REM}} = \mathcal{L}_{\text{GAN}}(E_d(G(\mathbf{Z}_\lambda)), 0) = -\frac{1}{b} \sum_{i=1}^{b} \log\left(1 - E_d\left(G\left(z_\lambda^{(i)}\right)\right)\right). \tag{8}$$

In our model, the adversarial process is simply described by a full backpropagation of error through $E_d$ and $G$ with a sign switch of weight changes in $G$.

In summary, each Wake–NREM–REM cycle consists of (1) reconstructing a mini-batch $x$ of images during Wake, (2) reconstructing a mini-batch of latent activities $\mathbf{Z} = E_z(\mathbf{X})$ during NREM with perturbation of $G(z)$, and (3) replaying $\mathbf{Z}$ convex combined with $\mathbf{Z}_{\text{old}}$ and noise from the $(n-1)$ th cycle. In PAD training, all losses are weighted equally and we did not use a schedule for $\mathcal{L}_{\text{KL}}$, as opposed to standard variational autoencoder training (***Kingma and Welling, 2013***). One training epoch is defined by the number of mini-batches necessary to cover the whole dataset. The evolution of losses with training epochs is shown in ***Appendix 1—figure 1*** and ***Appendix 1—figure 2***. The whole training procedure is summarized in the pseudo-code implemented in the section 'Algorithm 1:Training procedure'.

## Image occlusion

Following previous work (***Zeiler and Fergus, 2013***), gray squares of various sizes are applied along the image with a certain probability (***Figure 9***). For each mini-batch, a probability and square size were randomly picked between 0 and 1, and 1–8, respectively. We divide the image into patches of the given size and replace each patch with a constant value (here, 0) according to the defined probability.

## Evaluation
### Training of linear readout
A linear classifier is trained on top of latent features $\mathbf{Z} = E_z(\mathbf{X})$, with $\mathbf{Z} \in \mathbb{R}^{N \times 256}$, where $N$ is the number of training dataset images. A latent feature $z \in \mathbb{R}^{256}$ is projected via a weight matrix $W \in \mathbb{R}^{10 \times 256}$ to the label neurons to obtain the vector $y = Wz$.

This weight matrix is trained in a supervised fashion by using a multiclass cross-entropy loss. For a feature $z$ labeled with a target class $t \in \{0, 1, .., 9\}$, the per-sample classification loss is given by

$$\mathcal{L}^C(z, t; W) = -\log p_W(Y = t \mid z) . \tag{9}$$

Here, $p_{\mathbf{W}}$ is the conditional probability of the classifier defined by the linear projection and the softmax function

$$p_W(Y = t \mid z) = \frac{e^{y_t}}{\sum_{i=0}^{9} e^{y_i}} . \tag{10}$$

The classifier is trained by mini-batch ($b = 64$) stochastic gradient descent on the loss $\mathcal{L}^C$ with a learning rate $\eta = 0.2$ for 20 epochs using the whole training dataset.

## Linear separability

Following previous work (*Hjelm et al., 2019*), we define linear separability as the classification accuracy of the trained classifier on inferred latent activities $E_z(X_{test})$ from a separate test dataset $X_{test}$. Given a latent feature $z$, class prediction is made by picking the index of the maximal activity in the vector $y$. We ran several simulations for four different initial parameters of $E$ and $G$ and report the average test accuracy and standard error of the mean (SEM) over trials. To evaluate performance on occluded data, we applied random square occlusion masks on each sample from $X_{test}$ for a fixed probability of occlusion and square size. We report only results for occlusions of size 4, after observing similar results with other square sizes.

## PCA visualization

To visualize the 256-dimensional latent representation $E_z(x)$ of the trained model, we used the PCA reduction algorithm (*Jolliffe and Cadima, 2016*). We project the latent representations to the first two principal components.

## Latent space organization metrics

Intra-class distance is computed by randomly picking 1000 pairs of images of the same class, projecting them to the encoder latent space $z$ and computing their Euclidean distance. This process is repeated over the 10 classes in order to obtain the average over 10 classes. Similarly, inter-class distance is computed by randomly picking 10,000 pairs of images of different classes, projecting them to the encoder latent space $z$ and computing their Euclidean distance. The ratio of intra- and inter-class distance is obtained by dividing the mean intra-class distance by the mean inter-class distance. Clean-occluded distance is computed by randomly picking 10,000 pairs of nonoccluded/occluded images, projecting them to the encoder latent space, and computing their Euclidean distance. The ratio of clean-occluded and inter-class distance is obtained by dividing the clean-occluded distance by the mean inter-class distance. We performed this analysis for several different trained networks with different initial conditions and report the mean ratios and SEM over trials.

## Fréchet inception distance

Following *Heusel et al., 2018*, FID is computed by comparing the statistics of generated (NREM or REM) samples to real images from the training dataset projected through an Inception v3 network pre-trained on ImageNet

$$\text{FID} = \|\mu_{real} - \mu_{gen}\|^2 + \text{Tr}(\Sigma_{real} + \Sigma_{gen} - 2(\Sigma_{real}\Sigma_{gen})^{1/2}) \tag{11}$$

where $\mu$ and $\Sigma$ represent the empirical mean and covariance of the 2048-dimensional activations of the Inception v3 pool3 layer for 10,000 pairs of data samples and generated images. Results represent mean FID and SEM FID over four different trained networks with different initializations.

## Modifications specific to pathological models

To evaluate the differential effects of each phase, we removed NREM and/or REM phases from training (*Figures 4–6*). For instance, for the condition w/o NREM, the network is never trained with NREM.

A few adjustments were empirically observed to be necessary in order to obtain a fair comparison between each condition. When removing the REM phase during training, we observed a decrease of linear separability after some ($gt_{25}$) epochs. We suspect that this decrease is a result of overfitting due to unconstrained autoencoding objective of $E$ and $G$. Models trained without REM hence would not provide a good baseline to reveal the effect of adversarial dreaming on linear separability. For models without the REM phase, we hence added a vector of Gaussian noise $\epsilon \sim \mathcal{N}(0, 0.5 \cdot I)$ to the encoded activities $E_z(X)$ of dimension $n_z$ before feeding them to the generator. Thus, *Equation 2* becomes

$$\mathcal{L}_{img} = \frac{1}{b}\sum_{i=1}^{b} \| x^{(i)} - G\left(E_z(x^{(i)}) + \epsilon\right) \|^2 , \tag{12}$$

which stabilizes linear separability of latent activities around its maximal value for both CIFAR-10 and SVHN datasets until the end of training.

Furthermore, we observed that the NREM phase alters linear performance in the absence of REM (w/o REM condition). To overcome this issue, we reduced the effect of NREM by scaling down its loss with a factor of 0.5. This enabled to benefit from NREM (recognition under image occlusion) without altering linear separability on full images.

## Acknowledgements

This work has received funding from the European Union 7th Framework Programme under grant agreement 604102 (HBP), the Horizon 2020 Framework Programme under grant agreements 720270, 785907, and 945539 (HBP), the Swiss National Science Foundation (SNSF, Sinergia grant CRSII5-180316), the Interfaculty Research Cooperation (IRC) 'Decoding Sleep' of the University of Bern, and the Manfred Stärk Foundation. We thank the IRC collaborators Paolo Favaro for inspiring discussions on related methods in AI and deep learning, and Antoine Adamantidis and Christoph Nissen for helpful discussions on REM/NREM sleep phenomena in mice and humans.

## Additional information

### Funding

| Funder | Grant reference number | Author |
| --- | --- | --- |
| European Commission | 604102 | Mihai A Petrovici<br>Walter Senn<br>Jakob Jordan |
| European Commission | 720270 | Mihai A Petrovici<br>Walter Senn<br>Jakob Jordan |
| European Commission | 785907 | Mihai A Petrovici<br>Walter Senn<br>Jakob Jordan |
| European Commission | 945539 | Mihai A Petrovici<br>Walter Senn<br>Jakob Jordan |
| Universität Bern | Interfaculty Research Cooperation 'Decoding Sleep' | Nicolas Deperrois<br>Walter Senn |
| Universität Heidelberg | Manfred Stärk Foundation | Mihai A Petrovici |
| Swiss National Science Foundation | Sinergia Grant CRSII5-180316 | Walter Senn |

The funders had no role in study design, data collection and interpretation, or the decision to submit the work for publication.

### Author contributions

Nicolas Deperrois, Conceptualization, Data curation, Formal analysis, Investigation, Methodology, Software, Validation, Visualization, Writing – original draft, Writing – review and editing; Mihai A Petrovici, Conceptualization, Supervision, Validation, Visualization, Writing – review and editing; Walter Senn, Conceptualization, Funding acquisition, Methodology, Project administration, Supervision, Validation, Writing – review and editing, Shared senior authorship; Jakob Jordan, Conceptualization, Formal analysis, Methodology, Supervision, Validation, Visualization, Writing – original draft, Writing – review and editing, Shared senior authorship

## Author ORCIDs

Nicolas Deperrois http://orcid.org/0000-0001-7178-1818
Mihai A Petrovici http://orcid.org/0000-0003-2632-0427
Walter Senn http://orcid.org/0000-0003-3622-0497
Jakob Jordan http://orcid.org/0000-0003-3438-5001

## Decision letter and Author response

Decision letter https://doi.org/10.7554/eLife.76384.sa1
Author response https://doi.org/10.7554/eLife.76384.sa2

# Additional files

## Supplementary files

• Transparent reporting form

## Data availability

The current manuscript is a computational study, so no data have been generated for this manuscript. Deep learning benchmark datasets (CIFAR-10 and SVHN) were used for the simulations. We published all code necessary to repeat our experiments in the following repository: https://github.com/NicoZenith/PAD, (copy archived at swh:1:rev:0a2a4ad4da69f2f8c53fd5ee96e895c72aeb9f26).

The following previously published datasets were used:

| Author(s) | Year | Dataset title | Dataset URL | Database and Identifier |
|---|---|---|---|---|
| Krizhevsky A | 2009 | CIFAR-10 | https://www.cs.toronto.edu/~kriz/cifar.html | Department of Computer Science, University of Toronto, ~kriz/cifar |
| Netzer Y, Wang T, Coates A, Bissacco A, Wu B, Ng AY | 2011 | SVHN | http://ufldl.stanford.edu/housenumbers/ | Computer Science Department, Stanford University, housenumbers |

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

## Appendix 1

### Training losses for full and pathological models

In the following, we report the measured losses over training for the various different pathological conditions. $\mathcal{L}_{\text{img}}$ and $\mathcal{L}_{\text{KL}}$ are optimized for each condition and systematically decrease with learning, while $\mathcal{L}_{\text{NREM}}$ is significantly reduced in models with NREM (*Appendix 1—figure 1*, *Appendix 1—figure 2*). Its initial increase in the models with REM is explained by its competitive optimization with the GAN losses. Generator loss $\mathcal{L}_{\text{fake}} = \mathcal{L}_{\text{REM}}$ and discriminator loss $\mathcal{L}_{\text{real}} + \mathcal{L}_{\text{fake}}$ are only optimized in models with REM, showing a progressive decrease of the discriminator loss in parallel with an increase of the generator loss, reflecting adversarial learning between the two streams.

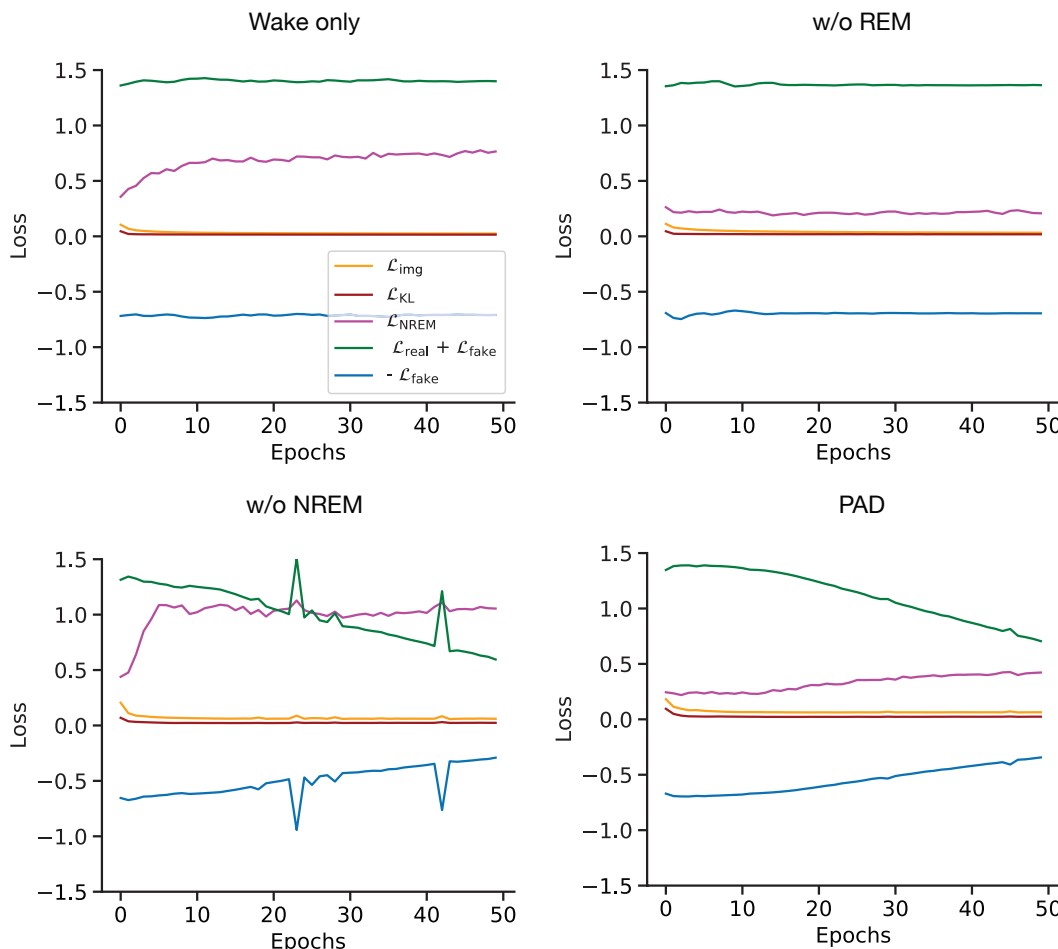

**Appendix 1—figure 1.** Training losses for the full and pathological models with the CIFAR-10 dataset. Evolution of training losses used to optimize $E$ and $G$ networks (see Materials and methods) over training epochs for the full and pathological models.

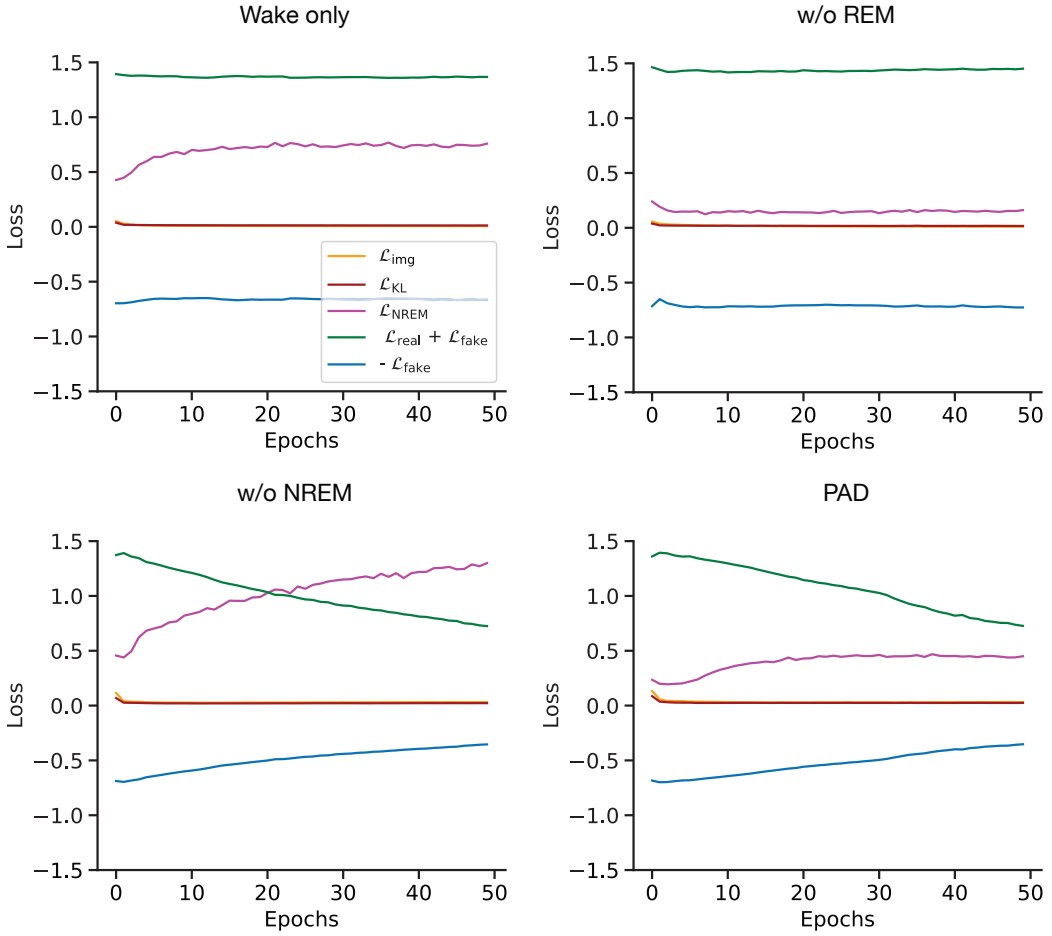

**Appendix 1—figure 2.** Training losses for the full and pathological models with the SVHN dataset.

## Linear classification performance

We report the mean and SEM of the final linear classification performance (epoch 50) on latent representations of the PAD and pathological models in *Appendix 1—table 1*.

**Appendix 1—table 1.** Final classification performance for the full model and all pathological conditions for unoccluded images.

Mean and standard error of the mean (SEM) over four different initial condition of linear separability of latent representations at the end of training (epoch 50) for perturbed and adversarial dreaming (PAD) and its pathological variants.

| Dataset | PAD | W/o memory mix | W/o REM | W/o NREM | Wake only |
|---------|-----|----------------|---------|----------|-----------|
| CIFAR-10 | 58.25 ± 0.70 | 53.87 ± 0.85 | 46.00 ± 0.43 | 58.00 ± 0.34 | 42.25 ± 0.54 |
| SVHN | 78.92 ± 0.40 | 60.87 ± 5.07 | 42.30 ± 1.51 | 73.25 ± 0.22 | 41.93 ± 0.65 |

REM: rapid eye movement; NREM: non-rapid eye movement.

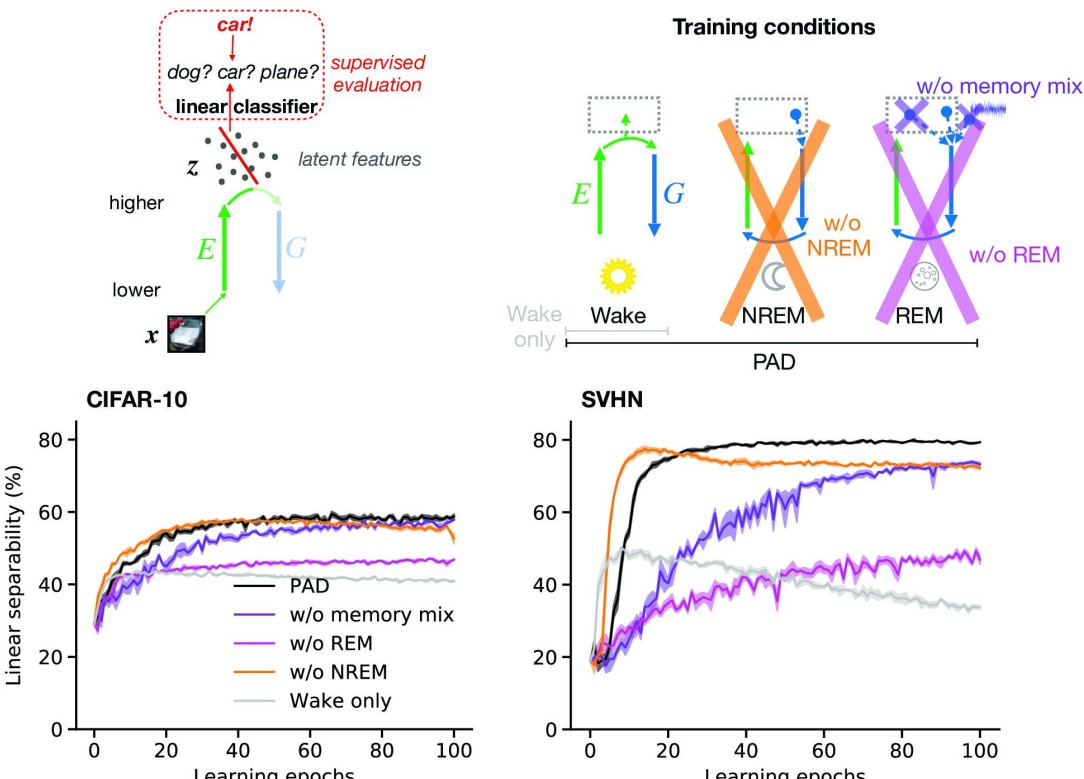

**Appendix 1—figure 3.** Linear classification performance for the full model and all pathological conditions. For details, see *Figure 4*.

We also report the linear classification performance for the full and pathological models over 100 epochs. Linear separability for the 'w/o REM' (*Appendix 1—figure 3c and d*, pink curves) and 'w/o memory mix' (*Appendix 1—figure 3d*, purple curve) conditions does not reach levels of the full model (*Appendix 1—figure 3c and d*, black curves) even after many training epochs. Furthermore, without NREM (*Appendix 1—figure 3c and d*, 'w/o NREM' and 'Wake only,' orange and gray curves), linear separability tends to decrease after many training epochs, suggesting that NREM helps to stabilize performance with training by preventing overfitting.

## Comparison of performance with REM driven by convex combination or noise

We report the linear classifier performance for PAD using different latent inputs to the generator. In the main text, we use a convex combination of mixed memories (being a convex combination of two different replayed latent vectors) and noise sampled from a Gaussian unit distribution (*Appendix 1—figure 4*, black). We here show the results when only random Gaussian noise is used (*Appendix 1—figure 4*, green) and when only a convex combination of memories is used (*Appendix 1—figure 4*, red). These different mixing strategies do not show a big difference in linear separability over training epochs.

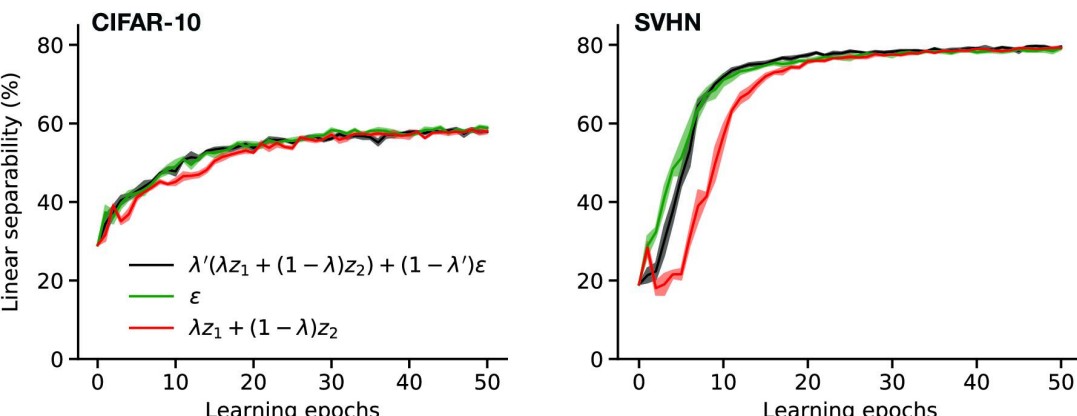

**Appendix 1—figure 4.** Linear classification performance for different mixing strategies during rapid eye movement (REM). Linear separability of latent representations with training epochs for perturbed and adversarial dreaming (PAD) trained with different REM phases: one driven by a convex combination of mixed memories and noise (black), one by pure noise (green), and one by mixed memories only (red). For details, see *Figure 4*.

## The order of sleep phases has no influence on the performance of the linear classifier

To investigate the role of the order of NREM and REM sleep phases, we consider a variation in which their order is reversed with respect to the model described in the main text. The performance of the linear classifier is not influenced by this change (*Appendix 1—figure 5*).

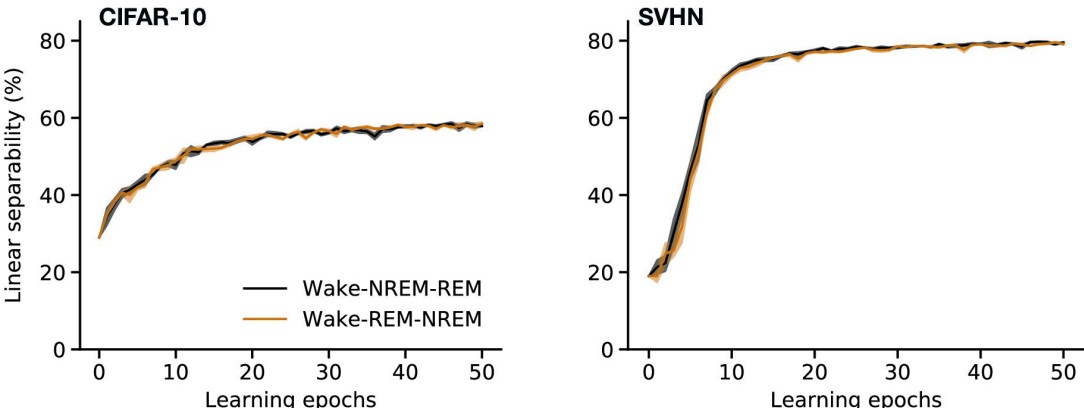

**Appendix 1—figure 5.** Linear classification performance for different order of sleep phases. Linear separability of latent representations with training epochs for perturbed and adversarial dreaming (PAD) trained when non-rapid eye movement (NREM) precedes rapid eye movement (REM) phase (Wake–NREM–REM, black) or when REM precedes NREM (Wake–REM–NREM, brown).

## Replaying multiple episodic memories during NREM sleep

While in the main text we considered NREM to use only a single episodic memory, here we report the results for a model in which also NREM uses multiple (here: two) episodic memories. In the full model (*Appendix 1—figure 6*, black curves, same data as in *Figure 5c and d*), NREM uses a single stored latent representation. Here, we additionally consider an additional model in which these representations are obtained from a convex combination of mixed memories and spontaneous cortical activity. The better performance of a single replay suggests that replay from single episodic memories as postulated to occur during NREM sleep is more efficient to robustify latent representations against input perturbations.

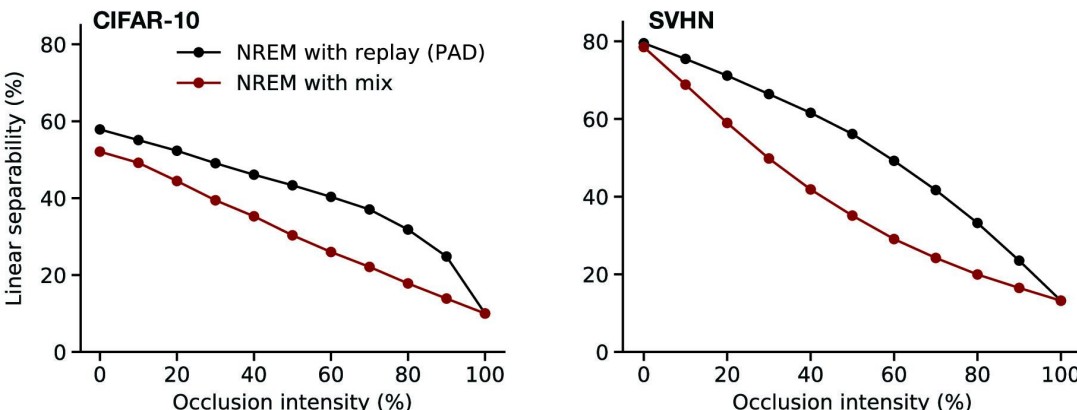

**Appendix 1—figure 6.** Importance of replaying single hippocampal memories during non-rapid eye movement (NREM). Linear separability of latent representations at the end of learning with occlusion intensity for a model trained with all phases.

