## [Editor Report]

This paper presents a generative adversarial network-inspired model of how learning during wakefulness, non-rapid eye movement (NREM), and REM sleep work together to facilitate the emergence of object category representations. The model is impressive in its ability to shape representations based on internally generated activity that does not directly recapitulate prior experience, and has properties that correspond to replay and dreams in NREM and REM sleep. The model makes predictions that can be tested in sleep experiments in humans and animals.

---

## [Decision Letter]

**Decision letter after peer review:**

[Editors’ note: the authors submitted for reconsideration following the decision after peer review. What follows is the decision letter after the first round of review.]

Thank you for submitting the paper "Memory semantization through perturbed and adversarial dreaming" for consideration by *eLife*. Your article has been reviewed by 3 peer reviewers, one of whom is a member of our Board of Reviewing Editors, and the evaluation has been overseen by a Senior Editor. The following individuals involved in review of your submission have agreed to reveal their identity: Blake A Richards (Reviewer #2); Ari Benjamin (Reviewer #3).

After consultation with the reviewers, we have decided on a rejection of the current manuscript with the possibility of resubmission of a substantially revised new manuscript. All reviewers felt that the model was interesting and impressive. The main issues that led us to this decision were a use of semanticization that seemed too different from its use in the cognitive neuroscience literature, making it unclear how to align the model and its predictions with the empirical literature, and lack of theory or analysis to make it clear why the model works. We felt that a version of the paper that addressed these points would essentially be a new paper, as it involves restructuring the framing and discussion, a new title, and substantial new analysis/theory. This comment from Reviewer 3 is the crux of the theory issue: "If the objective of GANs and VAEs are themselves insufficient for representation learning, absent architecture, why would their combination here avoid that problem? " If you decide to resubmit to *eLife*, the paper will likely go back to the same editors and reviewers.

*Reviewer #1:*

This paper presents a model inspired by generative adversarial networks that shows how different forms of nonveridical generative replay can lead to meaningful changes in representations. The authors simulated three cycling states––wakeful learning, NREM sleep, and REM sleep––as optimizing different learning objectives. The model consists of a feedforward pathway that learns to map low–level sensory inputs to high–level latent representations and to discriminate between external stimuli and internally simulated activity, and a feedback pathway that learns to reconstruct low–level patterns from high–level representations. During wakefulness, the model learns to reconstruct the sensory inputs it receives, while storing their associated latent vectors into a memory buffer representing the hippocampus. During NREM, low–level patterns are generated from individual memory traces in the hippocampus. The feedforward pathway processes occluded versions of these patterns and learns to reconstruct the original memory traces. In REM, patterns are generated based on random combinations of hippocampal memories. The feedforward pathway learns to avoid labeling these patterns as external inputs and the feedback pathway tries to trick the feedforward pathway into recognizing them as external stimuli. After training with several cycles of these three learning phases on naturalistic images, the model develops more linearly separable latent representations for different categories, which the authors interpret as evidence for semanticization.

There are several aspects of this model that are very interesting and impressive. It is able to reorganize internal memory representations in a meaningful way based only on internally generated activity. The demonstrations that occlusions in NREM work best on single episodes and that REM works best when using combined episodes have an intriguing correspondence to known properties of replay and dreams in these sleep stages. As detailed below, there are other aspects of the model that seem less consistent with known properties of sleep, and it is unclear whether the performance metrics demonstrate "semanticization" as the term is typically used in the literature.

1. One aspect of the model that stands out as being in tension with neural data is that cortical activity during REM is known to be less driven by the hippocampus than during NREM, due to fluctuations in acetylcholine. In the model, the hippocampus is the driver of replay throughout the model in both states.

2. The model predicts that internally generated patterns should become more similar to actual inputs over time, but REM dreams appear to be persistently bizarre, and in rodents there seems to be a decrease in veridical NREM replay over time after a new experience.

3. The use of linear separability as an index of semanticization seems in tension with the literature on semantics in a few ways. Areas of visual cortex that are responsive to certain categories of objects are not typically thought to be processing the semantic structure of those objects, in the way that higher level areas (e.g. anterior temporal lobes) do. Semanticization has the connotation of stripping episodes of their details and source information and representing the structure across experiences within the relevant domain, often in a modality–independent way. It is not clear that a simple separation between visual categories captures these senses of the word.

4. It is unclear when the predictions of the model apply to a night of sleep vs. several nights vs. hundreds or thousands (a developmental timescale). For example, the authors propose depriving subjects of REM sleep and testing the ability to form false memories. Putting aside the difficulty of REM deprivation, it is unclear how many nights of deprivation would be required to test the predictions of the model, especially because REM does not seem to be beneficial during the first few learning epochs (Figure 4).

Recommendations for the authors

1. McClelland, McNaughton, and O'Reilly 1995 is cited as a standard consolidation theory, but it belongs in the transformation theory category. The hippocampal memories in that model do not tend to retain their episodic character when consolidated in neocortex.

2. Norman, Newman, and Perotte 2005, Neural Networks, had a neural network model of REM that deserves discussion here.

3. The authors might consider simulations demonstrating to what extent alternation between sleep stages is needed and simulations demonstrating whether the order of replay matters – does the model behave differently if REM precedes NREM?

4. My understanding is that for analyses in Figure 5 the test dataset consists of occluded versions of training images. Does linear separability increase for occluded versions of images that are not presented during training?

5. Memories are linearly combined to guide reconstruction during REM. It could be useful to demonstrate that this does better than random activity patterns (patterns not based on memories).

*Reviewer #2:*

In this paper, Deperrois et al. develop a neural network model of unsupervised learning that uses three distinct training phases, corresponding to wakefulness, NREM sleep, and REM sleep. These phases are respectively, used to train for reconstructing inputs (and recognizing them as real), representing perturbed sensory inputs similar to non–perturbed sensory inputs, and recognizing internally generated inputs created from mixing stored memories. They show that this model can learn decent semantic concepts that are robust to perturbations, and they use ablation studies to examine the contribution of each phase to these abilities.

Overall, I really enjoyed this paper and I think it is fantastic. Its major strengths are its originality, its clarity, and its well–designed ablation studies. The authors have developed a model unlike any other in this area, and they have given the reader sufficient data to understand its design and how it works. I believe this paper will be important for researchers in the area of memory consolidation to consider. Moreover, the model makes interesting empirical predictions that can and should be tested.

The weaknesses of the paper are as follows:

1) It is odd that eliminating the NREM phase didn't have much of an impact on the accuracy for non–occluded images (Figure 5). My guess: classification on non–occluded images would drop more with the removal of NREM if the authors had used more perturbations than just occlusion, e.g. like those used in SimCLR. Though these additional experiments do not need to be included for the paper to be publishable, per se, I do think they should be considered by the authors (or other researchers) for future studies. This is particularly so because, as it stands, the results suggest that the NREM phase is merely helping the system to be better at recognizing occluded images, which is a wee bit trivial/obvious given that the NREM phase is literally training on occluded images. All that being said, Figure 6e seems to suggest that NREM does help with separating inter–class distances. So, I am left a little confused as to what the actual result is on this matter. The authors only discuss these issues briefly on lines 393–397, and this really could be expanded.

2) I do not see any reason to run z through the linear classifier weights before performing t–SNE. Moreover, I am concerned that this ends up just being equivalent to an alternative means of visualizing classification accuracy. First, t–SNE should be able to identify these clusters from z itself, and there is essentially no logic provided as to why it wouldn't be able to do this–after all, this is what t–SNE was designed to do. Second, the linear projection of z with the classifier weight will necessarily correspond to a projection of the z vectors that increases the separation between classes. So, really, what we're visualizing here is how well that linear projection separates the classes. But that is already measured by classification accuracy. As such, I don't see what this analysis does beyond the existing data on classification accuracy. I think the authors should have performed t–SNE on the z vectors directly. If the authors are determined not to do this, they should provide much better logic explaining why this is not an appropriate analysis. To reiterate: t–SNE is designed for this purpose and has been used like this in many other publications!

3) In the discussion on potential mechanisms for implementing the credit assignment proposed here, the authors only mention one potential solution when there are literally dozens of papers on biologically realistic credit assignment in recent years. Lillicrap et al. (2020) and Whittington and Bogacz (2019) both provide reviews of these papers. Plus, Payeur et al. (2021) provide an exhaustive table in their supplementary material listing the different solutions on offer and their properties. The authors should note that there are a multitude of potential solutions, not just one, and reference at least some of these.

Recommendations for the authors

1) It is probably worth noting/mentioning that most people report having dreams with completely novel/surreal elements that can be wholly different from their past experiences (e.g. flying), suggesting that not all dreams are a result of rearranging fragments from stored episodic memories. The authors should discuss this and recognize it as a potential limitation of the model.

2) The perturbed dreaming phase is highly reminiscent of existing self–supervised models from machine learning (e.g. SimCLR, BarlowTwins, etc.), since it is essentially training the feedforward network to match perturbed/transformed versions of the same images to the same latent state as each other. For sake of providing the reader with more intuition about what is happening in the model, the authors should expand the discussion of these links.

3) A few typos to fix:

– Line 30: organisms –> organism's

– Line 47: sleep state –> sleep states

– Line 341: Our NREM phase does not require to store raw sensory inputs… –> Our NREM phase does not require the storage of raw sensory inputs…

4) Figure 6e and f are confusing and need to be improved. First, it is unclear what the two different bars for each training regime represent. Second, the y–axes don't make it clear that this is the ratio of intra–to–inter class distances, and the legend has to be referred to for that, which is not helpful for clarity.

5) To be completely candid with the authors, Figure 7b is very confusing and not terribly helpful for the reader. I understand that this is a sketch of the authors' current thinking on how their PAD model could relate to cortical circuits, but making concrete sense of exactly what is being proposed is nigh impossible. I think the authors should consider removing this panel and simply noting in the text that there are potential biological mechanisms to make the PAD model feasible. As it stands, Figure 7b takes a strong, clear paper and ends it on a very confusing note…

6) In equation 1, are all three losses really weighted equally over all of training? I'm surprised that the KLD term isn't given a schedule. This is common with VAE models and can help with training.

7) In section 4.4.4 and 4.4.5 the numbers use a single quote to denote the thousands decimal, but that's a mistake: it should be a comma, e.g. 10,000 not 10'000.

8) Figure 10 and section 6.1: L_latent is never defined. What is it? Is that what equation 12 was supposed to define (which would make sense, given that equation 2 already defined L_img). Also, why does it increase during training? Similarly, L_fake is never defined.

*Reviewer #3:*

The proposal that the brain learns adversarially during sleep stages is fascinating. The authors propose that not only does feedback to the earliest areas form a generative model of those areas, but that also feedforward activity carries the dual interpretation of a discriminator. (This proposal aligns with that of Gershman (2019) https://www.frontiersin.org/articles/10.3389/frai.2019.00018/, which should be cited here). If it could be shown that this is indeed what the brain does the impact would be tremendous. However, the evidence presented in the manuscript does not yet make a strong case.

The paper focuses primarily on modeling semantization, and this is defined as the degree to which object categories can be linearly decoded from the top layer of an encoding/decoding neural network. It is worth noting that other communities might call this a model of 'unsupervised representation learning' rather than semantization during memory consolidation. But is linear decodability of object categories an equivalent concept to the semantization of episodic memory? This seems to miss much about memory consolidation.

The focus on decodability is also problematic in part because it's not clear what about the model leads to it. In the ML community, it is known that the objectives of generative modeling and autoencoding are by themselves insufficient to provide "good" representations measured by linear decodability of supervised labels. (For arguments why, see https://arxiv.org/pdf/1711.00464.pdf and https://arxiv.org/abs/1907.13625 for autoencoders and https://www.inference.vc/maximum–likelihood–for–representation–learning–2/ for GANs). If such a system empirically learns untangled representations of categories, it is because the network architecture or prior distribution over latents is constraining in some way. The authors claim that "generating new, virtual sensory inputs via adversarial dreaming during REM sleep is essential for extracting semantic concepts" (line 14–15, also 221–222). If the objective of GANs and VAEs are themselves insufficient for representation learning, absent architecture, why would their combination here avoid that problem? For example, is the DCGAN–like architecture crucial? This is possible, but only one architecture was tested. (It is also concerning that the linear decodability of representations in DCGANs can be much higher than reported here; some component of the model is deteriorating, rather than giving, this quality. See Radford et al. (2014)). What about the REM stage in particular is necessary – for example, does it work when randomly sampling from the prior over Z or just convex combinations? Overall, from the computational perspective, I don't think it is yet supported that this objective function necessarily leads to learning untangled, semantic representations from which labels are linearly decodable.

Linear decoding aside, is this a good model of neural physiology and plasticity? It's a promising direction, and I like the discussion of NREM and REM. However, for a model this radical to be convincing I think much more attention should be paid to what this would look like biologically. Some specific questions stand out:

– I find it concerning that the generative model is only over the low–level inputs, e.g. V1 (or do the authors believe it would be primary thalamus?). In the predictive processing literature, it is generally assumed that *at every layer* feedback forms an effective generative model of that layer. In the hierarchical model here, there is no relation between the intermediate activations in the feedforward path to those in the feedback path. This prevents the integration of top–down information in intermediate sensory areas and makes the model unrealistic.

– What neurobiological system do the authors propose implements the output discriminator? If there are no obvious candidates, what would it look like, and what sorts of experiments could identify it?

– What consequences would the re–use of the feedforward model as a discriminator have for sensory physiology? This is a rather radical change to traditional models of forward sensory processing.

– The proposed experiments would test if sleep stages are involved in learning, but wouldn't implicate any adversarial learning. For example, the proposal to interrupt REM sleep would not dissociate this proposal from any other in which REM sleep is involved in sensory learning.

– I think an article modeling consolidation should be situated in hippocampal modeling. Yet here the hippocampus is modeled simply as a RAM memory bank, and the bulk of modeling decisions are about cortical perceptual streams. If the proposal is that this is what the hippocampus effectively does, it would be nice to have a mechanistic discussion as to how the hippocampus might linearly interpolate between two memory traces during the NREM stage. In general, what would this predict a hippocampal physiologist would see?

– Many related algorithms are dismissed in lines 380–381. I'm not sure what optimization tricks have been removed. Perhaps the authors could explain what was removed and why this makes PAD biologically plausible. In my opinion many of these are comparable.

I love the originality of this work. Yet to be taken seriously I think it needs to be much more firmly rooted in experimental findings and predictions. A review/perspective format with demonstrative simulations could be more appropriate.

In my opinion the focus on semantization/ linear decodability is a cherry on top of the main proposal, which is the adversarial framework for sleep stages. Given my reservations about the decodability aspects I think it may be a stronger paper if the framing shifts to focus on sleep physiology and unsupervised learning.

Miscellaneous comments.

– Is it spelled semantization or semanticization? The latter appears to be in more common use.

– I found the tSNE plots not particularly useful. tSNE is nonlinear so it is not a measure of linear category untangling. Please say more about what exactly this measure means, and report the perplexity parameter and how it was chosen.

– The authors should be aware of recent failures to replicate the Berkes (2011) result: https://elifesciences.org/articles/61942

Finally, some citations that I think could be mentioned:

Previous proposals that the brain may learn adversarially:

– Gershman, Samuel J. "The generative adversarial brain." Frontiers in Artificial Intelligence 2 (2019): 18.

– https://arxiv.org/abs/2006.10811 (full disclosure, a work of my own)

Work in the ML community in which the encoder is also a discriminator:

– Brock, Andrew, Lim, Theodore, Ritchie, James M, and Weston, Nick. Neural photo editing with introspective adversarial networks. arXiv preprint arXiv:1609.07093, 2016.

– Ulyanov, Dmitry, Vedaldi, Andrea, and Lempitsky, Victor. It takes (only) two: Adversarial generatorencoder networks. In Thirty–Second AAAI Conference on Artificial Intelligence, 2018.

– Huang, Huaibo, He, Ran, Sun, Zhenan, Tan, Tieniu, et al. Introvae: Introspective variational autoencoders for photographic image synthesis. In Advances in neural information processing systems, pp. 52–63, 2018.

– Munjal, Prateek, Paul, Akanksha, and Krishnan, Narayanan C. Implicit discriminator in variational autoencoder. arXiv preprint arXiv:1909.13062, 2019.

– Bang, Duhyeon, Kang, Seoungyoon, and Shim, Hyunjung. Discriminator feature–based inference by recycling the discriminator of gans. International Journal of Computer Vision, pp. 1–23, 2020.

[Editors’ note: further revisions were suggested prior to acceptance, as described below.]

Thank you for resubmitting your work entitled "Learning cortical representations through perturbed and adversarial dreaming" for further consideration by eLife. Your revised article has been evaluated by the same three reviewers.

The manuscript has been improved and the reviewers were overall positive, but they have one significant remaining request and one minor request for a caveat:

The significant request is to expand on how exactly the theory can be tested in experiments, with particular emphasis on diagnostic experiments that would allow us to rule out plausible alternatives. From Reviewer 3: "It takes a lot of thought to imagine how this particular hypothesis would surface in data and I don't think it should be left to the reader. More specifically, the paper still has no experimental predictions that could separate this idea from other similar possibilities involving generative models. The authors agreed in the response that they pose few predictions for the adversarial component, and instead only for "whether REM sleep is involved in cortical representation learning using a generative model." The anterior prefrontal cortex is now briefly mentioned as the top of the discriminator, but surely there would be a great deal of evidence in connectivity, power over plasticity, lesion studies, etc., that could confirm this. Re: "we interpret the reported novelty of REM dreams as strong existing evidence that this learning is based on adversarial principles rather than driven by reconstructions," (531-533): Here I disagree (though yes, not reconstructions). It is a strong hint that it is driven by an offline stage involving generative processes. This need not be adversarial. The Wake-Sleep algorithm, for example, also has a Sleep phase in which the hierarchical generative model generates samples via ancestral sampling from the top-level prior distribution. (Perhaps there is a misunderstanding regarding WS: the introduction currently dismisses the WS algorithm with the sentence, "these models explicitly try to reconstruct observed sensory inputs, while most dreams observed during REM sleep rarely reproduce past sensory experiences", lines 40-41. WS does try to reconstruct inputs during wake, but during sleep it 'fantasizes' randomly like a GAN.) Thus I still feel the paper does not offer tests by which we could know if the model were true."

The minor request is to acknowledge in the intro and the new discussion paragraph that the generative algorithm likely requires certain architectures & priors to deliver semantic representations.

Typo: "Moreover, removing NREM from training also increases [this] ratio." .

---

## [Author Response]

[Editors’ note: the authors resubmitted a revised version of the paper for consideration. What follows is the authors’ response to the first round of review.]

Reviewer #1:This paper presents a model inspired by generative adversarial networks that shows how different forms of nonveridical generative replay can lead to meaningful changes in representations. The authors simulated three cycling states––wakeful learning, NREM sleep, and REM sleep––as optimizing different learning objectives. The model consists of a feedforward pathway that learns to map low–level sensory inputs to high–level latent representations and to discriminate between external stimuli and internally simulated activity, and a feedback pathway that learns to reconstruct low–level patterns from high–level representations. During wakefulness, the model learns to reconstruct the sensory inputs it receives, while storing their associated latent vectors into a memory buffer representing the hippocampus. During NREM, low–level patterns are generated from individual memory traces in the hippocampus. The feedforward pathway processes occluded versions of these patterns and learns to reconstruct the original memory traces. In REM, patterns are generated based on random combinations of hippocampal memories. The feedforward pathway learns to avoid labeling these patterns as external inputs and the feedback pathway tries to trick the feedforward pathway into recognizing them as external stimuli. After training with several cycles of these three learning phases on naturalistic images, the model develops more linearly separable latent representations for different categories, which the authors interpret as evidence for semanticization.

We appreciate that the reviewer fully understood the mechanisms of our model and its aim. In the following, we reply point-by-point to the reviewer’s feedback.

There are several aspects of this model that are very interesting and impressive. It is able to reorganize internal memory representations in a meaningful way based only on internally generated activity. The demonstrations that occlusions in NREM work best on single episodes and that REM works best when using combined episodes have an intriguing correspondence to known properties of replay and dreams in these sleep stages. As detailed below, there are other aspects of the model that seem less consistent with known properties of sleep, and it is unclear whether the performance metrics demonstrate "semanticization" as the term is typically used in the literature.

We thank the reviewer for appreciating the link of computational properties of our model to known properties of sleep and dreaming. We agree that the model does not cover all aspects of semantization as typically used in the literature but that it rather suggests mechanisms for one aspect of semantization, the learning of useful cortical representations. We thus changed the framing of our manuscript by focusing on representation learning rather than semantization.

1. One aspect of the model that stands out as being in tension with neural data is that cortical activity during REM is known to be less driven by the hippocampus than during NREM, due to fluctuations in acetylcholine. In the model, the hippocampus is the driver of replay throughout the model in both states.

We agree with the reviewer that hippocampal influence is reduced during REM sleep (e.g., Lewis et al., 2018). Nevertheless, dreams in human patients with hippocampal damage were reported to be less frequent and less episodic-like in nature (Spano` et al., 2020), showing that the hippocampus is at least partially involved in dreaming. We have adapted our model to reflect this combination of reduced hippocampal drive and increased cortical drive during REM by using a combination of episodic memories from hippocampus with spontaneous cortical activity. The conclusions drawn from our model are not influenced by this change. This is expected, since the main purpose of interpolating memories as in our original model is to cover points in latent space that were not experienced by the generator during wakefulness. Hence, adding noise as in the revised model has only small influences on learning.

2. The model predicts that internally generated patterns should become more similar to actual inputs over time, but REM dreams appear to be persistently bizarre, and in rodents there seems to be a decrease in veridical NREM replay over time after a new experience.

Indeed, the model predicts that generated patterns become more similar to inputs over time. However, an important distinction is which aspects of the patterns become similar over time. We use the Fr´echet inception distance (FID) that leverages representations in the deep layers of a pre-trained neural network to measure similarity between dreams and external inputs. In particular, neural networks are known to focus on low-level properties of the input. Our model hence predicts that low-level properties of dreams become more structured, just as reported in the literature (Nir and Tononi, 2010). Capturing the “logical realism” of our generated neuronal activities most likely requires a more sophisticated evaluation metric and an extension of the model capable of generating temporal sequences of sensory stimulation. We discuss this point in the revised manuscript (section 3, lines 505-515).

3. The use of linear separability as an index of semanticization seems in tension with the literature on semantics in a few ways. Areas of visual cortex that are responsive to certain categories of objects are not typically thought to be processing the semantic structure of those objects, in the way that higher level areas (e.g. anterior temporal lobes) do. Semanticization has the connotation of stripping episodes of their details and source information and representing the structure across experiences within the relevant domain, often in a modality–independent way. It is not clear that a simple separation between visual categories captures these senses of the word.

We agree that our model addresses only one aspect of semantization, namely cortical representation learning. Please note that representation learning does capture some essential aspects of semantization as it leads to cortical activity which reflects semantic content of stimuli which are stripped of (some of) their episodic details. We revised the manuscript accordingly, especially the introduction, the discussion and the interpretation of our results. Furthermore, we agree that the extension to multiple modalities is interesting, and we expect our model to also be able to extract supra-modal information by introducing multiple sensory pathways, e.g., auditory in addition to visual. In the revised manuscript, we discuss this in section 3, lines 443-449.

4. It is unclear when the predictions of the model apply to a night of sleep vs. several nights vs. hundreds or thousands (a developmental timescale). For example, the authors propose depriving subjects of REM sleep and testing the ability to form false memories. Putting aside the difficulty of REM deprivation, it is unclear how many nights of deprivation would be required to test the predictions of the model, especially because REM does not seem to be beneficial during the first few learning epochs (Figure 4).

Indeed, our model aims at capturing the effects of REM over many nights, i.e., development. In the revised manuscript, we emphasize this point more strongly to reduce confusion in section 1, lines 67. We also adapted the experimental predictions (section 3, lines 517-525) to reflect this.

5. McClelland, McNaughton, and O'Reilly 1995 is cited as a standard consolidation theory, but it belongs in the transformation theory category. The hippocampal memories in that model do not tend to retain their episodic character when consolidated in neocortex.

We thank the reviewer for pointing this out. Due to the shift of the focus of the manuscript, we however do not discuss the differences between consolidation and transformation theories in detail any more.

6. Norman, Newman, and Perotte 2005, Neural Networks, had a neural network model of REM that deserves discussion here.

We thank the reviewer for the reference. Norman et al.’s model of REM and inhibitory oscillatory inhibition presents another potential role of REM in learning representations. We discussed this model in section 3, lines 410-417.

7. The authors might consider simulations demonstrating to what extent alternation between sleep stages is needed and simulations demonstrating whether the order of replay matters – does the model behave differently if REM precedes NREM?

This is an interesting point. We have run additional experiments to investigate the role of the order of sleep phases and found that it has negligible effect on our results. The independence of phases in our model may be due to the statistically similarity of training samples across training. In the revised manuscript we report these results in section 6.4. of the revised manuscript and discuss them in section 3 (lines 483-489).

8. My understanding is that for analyses in Figure 5 the test dataset consists of occluded versions of training images. Does linear separability increase for occluded versions of images that are not presented during training?

Unfortunately, we did not make this clear in the original manuscript, thanks for pointing this out. We did present occluded versions of images from the test dataset, not shown during training. In the revised manuscript, we clarify this point (section 2.5, lines 263-264).

9. Memories are linearly combined to guide reconstruction during REM. It could be useful to demonstrate that this does better than random activity patterns (patterns not based on memories).

We thank the reviewer for this suggestion. This variant would be similar to the original formulation of GANs (Goodfellow et al., 2014). Similar to our approach, other work in machine learning used convex combinations of encoded vectors to train autoencoders and produced good representations (Beckham et al., 2019; Berthelot et al., 2018). In the revised manuscript, we use a convex combination of mixed memories and random activity patterns mimicking spontaneous activity in cortex. While in the main manuscript we choose equal contributions of episodic memories and spontaneous background activity for simplicity, we include additional experiments in the supplements in which we investigate the extremes, i.e., only combinations of episodic memories or only noise. These results show no significant difference between these versions. We hypothesize that this invariance arises in our model due the statistical similarity of training samples across all training epochs. We speculate, however, that for models which learn continuously a preferential replay of combinations of episodic memories encourages the formation of representations which are useful in the more recent context. Loosely speaking, using episodic memories biases the model to focus on what is relevant now rather than what was relevant at some point. We have included a discussion of this point in the revised manuscript in section 3, lines 470-475.

Reviewer #2:In this paper, Deperrois et al. develop a neural network model of unsupervised learning that uses three distinct training phases, corresponding to wakefulness, NREM sleep, and REM sleep. These phases are respectively, used to train for reconstructing inputs (and recognizing them as real), representing perturbed sensory inputs similar to non–perturbed sensory inputs, and recognizing internally generated inputs created from mixing stored memories. They show that this model can learn decent semantic concepts that are robust to perturbations, and they use ablation studies to examine the contribution of each phase to these abilities.Overall, I really enjoyed this paper and I think it is fantastic. Its major strengths are its originality, its clarity, and its well–designed ablation studies. The authors have developed a model unlike any other in this area, and they have given the reader sufficient data to understand its design and how it works. I believe this paper will be important for researchers in the area of memory consolidation to consider. Moreover, the model makes interesting empirical predictions that can and should be tested.

We thank the reviewer for this positive feedback and for appreciating the novelty of our model. In the following, we reply point-by-point to the reviewer’s feedback.

The weaknesses of the paper are as follows:1) It is odd that eliminating the NREM phase didn't have much of an impact on the accuracy for non–occluded images (Figure 5). My guess: classification on non–occluded images would drop more with the removal of NREM if the authors had used more perturbations than just occlusion, e.g. like those used in SimCLR. Though these additional experiments do not need to be included for the paper to be publishable, per se, I do think they should be considered by the authors (or other researchers) for future studies. This is particularly so because, as it stands, the results suggest that the NREM phase is merely helping the system to be better at recognizing occluded images, which is a wee bit trivial/obvious given that the NREM phase is literally training on occluded images. All that being said, Figure 6e seems to suggest that NREM does help with separating inter–class distances. So, I am left a little confused as to what the actual result is on this matter. The authors only discuss these issues briefly on lines 393–397, and this really could be expanded.

We agree with the reviewer that our NREM phase is reminiscent of one phase of contrastive learning in which representations of different views of the same image are pulled together (Chen et al., 2020; Liu et al., 2021; Zbontar et al., 2021). However, our training is missing the negative phase in which representations of different images are pushed apart. We thus believe that the NREM phase in our model should rather be interpreted as a form of data augmentation mainly useful for robustifying against occlusions (as the reviewer correctly points out). We agree that including such a negative phase to perform a form of contrastive learning is a very promising future research direction. In the revised manuscript we discuss the link between our model and contrastive learning in section 3 (lines 459-466).

2) I do not see any reason to run z through the linear classifier weights before performing t–SNE. Moreover, I am concerned that this ends up just being equivalent to an alternative means of visualizing classification accuracy. First, t–SNE should be able to identify these clusters from z itself, and there is essentially no logic provided as to why it wouldn't be able to do this–after all, this is what t–SNE was designed to do. Second, the linear projection of z with the classifier weight will necessarily correspond to a projection of the z vectors that increases the separation between classes. So, really, what we're visualizing here is how well that linear projection separates the classes. But that is already measured by classification accuracy. As such, I don't see what this analysis does beyond the existing data on classification accuracy. I think the authors should have performed t–SNE on the z vectors directly. If the authors are determined not to do this, they should provide much better logic explaining why this is not an appropriate analysis. To reiterate: t–SNE is designed for this purpose and has been used like this in many other publications!

We agree with the reviewer that in principle t-SNE does not require any additional mapping as it should be able to identify these clusters autonomously. We initially used the linear classifier projection as we observed that clustered were easier discovered by t-SNE. We agree however that this analysis does not provide any significant new insights. To remove interpretation difficulties easily arising from t-SNE projections (Wattenberg et al., 2016) and make this analysis more accessible, we have replaced it with Principal Component Analysis (Jolliffe and Cadima, 2016) directly on the latent space to visualize its organization. The results obtained from PCA are very similar to those of tSNE.

3) In the discussion on potential mechanisms for implementing the credit assignment proposed here, the authors only mention one potential solution when there are literally dozens of papers on biologically realistic credit assignment in recent years. Lillicrap et al. (2020) and Whittington and Bogacz (2019) both provide reviews of these papers. Plus, Payeur et al. (2021) provide an exhaustive table in their supplementary material listing the different solutions on offer and their properties. The authors should note that there are a multitude of potential solutions, not just one, and reference at least some of these.

We agree that modern neuroscience is in the fortunate position to have a multitude of suggestions for efficient bioplausible credit assignment. For simplicity and fairness, we now cite the reviews the reviewer mentioned (Lillicrap et al., 2020; Whittington and Bogacz, 2019) and the perspective paper from Richards et al. (2019) in section 3, lines 439.

1) It is probably worth noting/mentioning that most people report having dreams with completely novel/surreal elements that can be wholly different from their past experiences (e.g. flying), suggesting that not all dreams are a result of rearranging fragments from stored episodic memories. The authors should discuss this and recognize it as a potential limitation of the model.

We agree with the reviewer that the typical reports contain experiences which are very different from past experiences. In the revised manuscript, we consider not just combinations of episodic memories but in addition the influence of spontaneous cortical memory during REM, in part to reflect the reduced correlation between hippocampal and cortical activity during this sleep stage (Wierzynski et al., 2009). We hypothesize that such a combination of episodic memories with ongoing cortical activity may indeed support surreal experiences. Nevertheless, we believe that also the described surreal experience of flying can be interpreted purely as the combination of episodic memories. We also would like to point out that the decrease in FID score reported on our manuscript does *not* mean that the semantic content of dreams, as measured by reports of bizarreness (Mamelak and Hobson, 1989), becomes more plausible over development. Capturing the ”logical realism” of our generated neuronal activities most likely requires a more sophisticated evaluation metric and an extension of the model capable of generating temporal sequences of sensory stimulation. In the revised manuscript, we discuss these points in detail in section 3, lines 505-515.

2) The perturbed dreaming phase is highly reminiscent of existing self–supervised models from machine learning (e.g. SimCLR, BarlowTwins, etc.), since it is essentially training the feedforward network to match perturbed/transformed versions of the same images to the same latent state as each other. For sake of providing the reader with more intuition about what is happening in the model, the authors should expand the discussion of these links.

As discussed above, our current model only implements one phase of these self-supervised models. Nevertheless we agree that this link is important and in the revised manuscript we discuss the link with contrastive learning methods in (section 3 lines 459-466).

3) A few typos to fix:– Line 30: organisms –> organism's– Line 47: sleep state –> sleep states– Line 341: Our NREM phase does not require to store raw sensory inputs… –> Our NREM phase does not require the storage of raw sensory inputs…

We thank the reviewer to point out these typos which are fixed in the revised manuscript.

4) Figure 6e and f are confusing and need to be improved. First, it is unclear what the two different bars for each training regime represent. Second, the y–axes don't make it clear that this is the ratio of intra–to–inter class distances, and the legend has to be referred to for that, which is not helpful for clarity.

We changed Figure 6e and f to make them more accessible, changing the y-axis into ‘ratio of intra/inter distances’ and added the dataset names to the first bars.

5) To be completely candid with the authors, Figure 7b is very confusing and not terribly helpful for the reader. I understand that this is a sketch of the authors' current thinking on how their PAD model could relate to cortical circuits, but making concrete sense of exactly what is being proposed is nigh impossible. I think the authors should consider removing this panel and simply noting in the text that there are potential biological mechanisms to make the PAD model feasible. As it stands, Figure 7b takes a strong, clear paper and ends it on a very confusing note…

We agree that the presentation of Figure 7b was suboptimal and that this content may be more appropriate for a perspective paper. We thus removed Figure 7b from the revised manuscript.

6) In equation 1, are all three losses really weighted equally over all of training? I'm surprised that the KLD term isn't given a schedule. This is common with VAE models and can help with training.

Indeed, all three losses are weighted equally over training. The only exception is during WAKE+NREM training where we reduced the weight of the NREM loss to 0*.*5 (see also Methods), which lead to better performance of the linear readout. Our training procedure did not make use of a schedule for the KL loss.

7) In section 4.4.4 and 4.4.5 the numbers use a single quote to denote the thousands decimal, but that's a mistake: it should be a comma, e.g. 10,000 not 10'000.

We thank the reviewer for pointing this out, we corrected it accordingly.

8) Figure 10 and section 6.1: L_latent is never defined. What is it? Is that what equation 12 was supposed to define (which would make sense, given that equation 2 already defined L_img). Also, why does it increase during training? Similarly, L_fake is never defined.

Indeed, we did not define this properly in the original manuscript, we thank the reviewer for pointing this out and their attention to detail. L_latent_ was actually the L_NREM_ defined in Equation 6., L_fake_ is defined in section 6.1., and is actually equal to L_REM_. We have improved this exposition in the revised manuscript.

Reviewer #3:The proposal that the brain learns adversarially during sleep stages is fascinating.

We appreciate that the reviewer shares the same vision of the brain and their enthusiasm. In the following, we reply point-by-point to the reviewer’s feedback.

The authors propose that not only does feedback to the earliest areas form a generative model of those areas, but that also feedforward activity carries the dual interpretation of a discriminator. (This proposal aligns with that of Gershman (2019) https://www.frontiersin.org/articles/10.3389/frai.2019.00018/, which should be cited here). If it could be shown that this is indeed what the brain does the impact would be tremendous. However, the evidence presented in the manuscript does not yet make a strong case.

We thank the reviewer for the reference, in the revised manuscript we refer to Gershman (2019) in sections 2.6 (lines 334) and 3 (lines 418). However, please note that in contrast to our model Gershman mainly focuses on reality monitoring and does not propose a concrete implementation, only referring to adversarially learned inference (Dumoulin et al., 2017).

The paper focuses primarily on modeling semantization, and this is defined as the degree to which object categories can be linearly decoded from the top layer of an encoding/decoding neural network. It is worth noting that other communities might call this a model of 'unsupervised representation learning' rather than semantization during memory consolidation. But is linear decodability of object categories an equivalent concept to the semantization of episodic memory? This seems to miss much about memory consolidation.

We agree with the reviewer that our model does not capture the various facets of semantization discussed in the cognitive literature on memory consolidation. Rather, as the reviewer correctly points out, we address one component relevant for semantization, namely representation learning. In the revised manuscript, we hence sharpen the focus on cortical representation learning. Nevertheless, we are convinced that understanding representation learning is a key step towards understanding semantization as a whole.

The focus on decodability is also problematic in part because it's not clear what about the model leads to it. In the ML community, it is known that the objectives of generative modeling and autoencoding are by themselves insufficient to provide "good" representations measured by linear decodability of supervised labels. (For arguments why, see https://arxiv.org/pdf/1711.00464.pdf and https://arxiv.org/abs/1907.13625 for autoencoders and https://www.inference.vc/maximum–likelihood–for–representation–learning–2/ for GANs). If such a system empirically learns untangled representations of categories, it is because the network architecture or prior distribution over latents is constraining in some way. The authors claim that "generating new, virtual sensory inputs via adversarial dreaming during REM sleep is essential for extracting semantic concepts" (line 14–15, also 221–222). If the objective of GANs and VAEs are themselves insufficient for representation learning, absent architecture, why would their combination here avoid that problem? For example, is the DCGAN–like architecture crucial? This is possible, but only one architecture was tested. (It is also concerning that the linear decodability of representations in DCGANs can be much higher than reported here; some component of the model is deteriorating, rather than giving, this quality. See Radford et al. (2014)). What about the REM stage in particular is necessary – for example, does it work when randomly sampling from the prior over Z or just convex combinations? Overall, from the computational perspective, I don't think it is yet supported that this objective function necessarily leads to learning untangled, semantic representations from which labels are linearly decodable.

We thank the reviewer for providing these references and we agree that theoretically it is not well understood how disentangled representations are learned from objectives which do not directly encourage them, i.e., reconstruction and adversarial losses. Nevertheless, as the reviewer points out, there exists a plethora of evidence that empirically these systems do learn useful representations. In line with previous arguments, we believe that it is the combination of objectives, architectural constraints, and latent priors which encourages the learning of disentangled representations (Tschannen et al., 2020).

In the revised manuscript we discuss these important theoretical points. Note however that we are focusing on leveraging empirically successful methods to explain computational and phenomenological properties of sleep and do not aim at an exhaustive theoretical evaluation of different representation learning techniques.

The linear decodability reported in the DCGANs paper (Radford et al., 2015) has been computed differently from our model case, explaining the much higher linear separability. In the revised manuscript, we discuss this point in section 3, lines 428-433.

Reflecting the reduced correlation between hippocampus and cortex during REM sleep (Wierzynski et al., 2009), we have modified the construction of latent states during REM in the revised manuscript to contain both a mixture of episodic memories and spontaneous cortical activity. While different mixing strategies could be used, we picked one that matches well with dream phenomenology. Extending the results of the manuscript, we show in the appendix (section 6.3) the performance with REM driven by convex combination or random Gaussian noise and report that there is no big difference. We speculate however that for models which learn continuously, a preferential replay of combinations of episodic memories encourages the formation of representations which are useful in the more recent context. We discuss this in the revised manuscript in section 3, lines 470-475.

Linear decoding aside, is this a good model of neural physiology and plasticity? It's a promising direction, and I like the discussion of NREM and REM. However for a model this radical to be convincing I think much more attention should be paid to what this would look like biologically. Some specific questions stand out:– I find it concerning that the generative model is only over the low–level inputs, e.g. V1 (or do the authors believe it would be primary thalamus?). In the predictive processing literature, it is generally assumed that *at every layer* feedback forms an effective generative model of that layer. In the hierarchical model here, there is no relation between the intermediate activations in the feedforward path to those in the feedback path. This prevents the integration of top–down information in intermediate sensory areas and makes the model unrealistic.

We thank the reviewer for bringing this point up. First, we would like to clarify that the generative pathway in our model generates activities across all layers during both NREM and REM sleep. Second, we agree that our implementation contrasts with the traditional view of the visual cortex where all bottom-up and top-down activities are merged at every layer. From a computational perspective of representation learning, such an architecture can be challenging to train, due to information shortcuts, e.g., V1 → V2 → V1, which would prevent information (at least during reconstruction learning) to propagate to higher areas (e.g., Inferior-Temporal cortex) where compressed representations should be learned. Naturally, this issue would also arise in predictive processing models (unless explicitly or implicitly prevented) as these information shortcuts are a property of the underlying graphical model and not of a particular implementation thereof.

Some strategies exist to achieve training of recurrent networks in the presence of such shortcuts, e.g., abandoning backpropagation training in favor of backpropagation through time (Spoerer et al., 2017), (greedy) layerwise training (Hinton, 2012), or using adversarial principles (Benjamin and Kording, 2021). We hypothesize that local recurrence and merging of the two information streams at intermediate layers are especially beneficial for processing of temporally-extended stimuli. Extending our model with local recurrence and combining it with appropriate training principles is an exciting avenue for future work.

Note however, that our model only describes the ”functional” connections (which are purely feedforward). In bioplausible implementations of backpropagation (Lillicrap et al., 2020; Whittington and Bogacz, 2019), additional connections would appear which are likely to be found in the biological substrate. In the revised manuscript, we discuss these points in section 2.6., lines 360-366.

– What neurobiological system do the authors propose implements the output discriminator? If there are no obvious candidates, what would it look like, and what sorts of experiments could identify it?

In the revised manuscript, we discuss the potential neurobiological systems to implement the output discriminator in section 2.6., lines 333-334, referring to the initial proposal from Gershman (2019) Indeed, the anterior prefrontal cortex has been shown to play a major role in reality monitoring by distinguishing internally from externally generated information (Simons et al., 2017).

– What consequences would the re–use of the feedforward model as a discriminator have for sensory physiology? This is a rather radical change to traditional models of forward sensory processing.

We agree with the reviewer that this is a break with traditional models of forward sensory processing which are focused on distinguishing the *content* of stimuli (e.g., object identity), whereas in our model the discriminator needs to distinguish the *source* of stimuli (“inside” vs. “outside”). However, it is commonly assumed that cortex performs many functions within sensory processing streams (DiCarlo et al., 2012). The distinction between externally or internally generated, while computationally new, is mechanistically not too different from the distinction of different object types, i.e., ultimately it is a classification based on stimulus properties. We have included a discussion of this in the revised manuscript in section 2.1, lines 125-131.

– The proposed experiments would test if sleep stages are involved in learning, but wouldn't implicate any adversarial learning. For example, the proposal to interrupt REM sleep would not dissociate this proposal from any other in which REM sleep is involved in sensory learning.

We agree with the reviewer that many predictions of our model address mainly the question of whether REM sleep is involved in cortical representation learning using a generative model. While we believe that experimentally testing this prediction would be a significant step forward, we would additionally like to point out that the reported novelty of REM dreams is a strong hint that this learning is not driven by reconstruction, but rather by adversarial principles. Otherwise, the similarity of dreams to waking experience should be higher. In addition, we make specific predictions for plasticity switching sign between wake and sleep phases, which should only be the case for adversarial learning. We discuss these points in section 3, lines 531-533 of the revised manuscript.

– I think an article modeling consolidation should be situated in hippocampal modeling. Yet here the hippocampus is modeled simply as a RAM memory bank, and the bulk of modeling decisions are about cortical perceptual streams. If the proposal is that this is what the hippocampus effectively does, it would be nice to have a mechanistic discussion as to how the hippocampus might linearly interpolate between two memory traces during the NREM stage. In general, what would this predict a hippocampal physiologist would see?

We agree that most sleep models focus on hippocampal replay. Yet semantic knowledge is represented in the neocortex, and sleep contributes to extracting semantic information from hippocampal memories in the cortex (Inostroza and Born, 2013). Hence, we believe that focusing on the neocortex and representation learning brings a missing computational principle to sleep models. Due to our focus, a simple hippocampal storage and replay mechanism is desirable and detailed hippocampal modeling would go beyond the scope of our manuscript.

Note that the combination of episodic memories does not need to take place in hippocampus, but rather it could occur in cortical neurons to which hippocampal neurons project to. Based on these ideas we would predict that one should observe a temporally close activation of different episodic memories in hippocampus during REM. In the revised manuscript, we discuss a potential hippocampus implementation, and the way memories would be combined during REM in section

2.6, lines 367-373.

– Many related algorithms are dismissed in lines 380–381. I'm not sure what optimization tricks have been removed. Perhaps the authors could explain what was removed and why this makes PAD biologically plausible. In my opinion many of these are comparable.

We thank the reviewer for pointing this out. We have improved the corresponding paragraph in the revised manuscript and now explicitly mention what was removed. Note however that we do not dismiss them entirely as computationally powerful and potentially implementable, rather we would like to highlight that even in the absence of these mechanisms, our model behaves well.

I love the originality of this work. Yet to be taken seriously I think it needs to be much more firmly rooted in experimental findings and predictions. A review/perspective format with demonstrative simulations could be more appropriate.

We appreciate the reviewers enthusiasm for our model. We however disagree that a review/perspective format would be better suited. From the construction of the model over the relation to brain physiology and cognitive theories to the competitive computational results, our manuscript clearly presents suitable material for an original research article and an advancement of the state of the art.

In my opinion the focus on semantization/ linear decodability is a cherry on top of the main proposal, which is the adversarial framework for sleep stages. Given my reservations about the decodability aspects I think it may be a stronger paper if the framing shifts to focus on sleep physiology and unsupervised learning.

We thank the reviewer for appreciating the novelty of our adversarial approach to sleep and dreams. However, we disagree that decodability, or in general representation learning is the cherry on top. On the contrary we are convinced that transforming sensory input into representations which are useful for a variety of downstream tasks is a core component of cortical information processing. The linear readout here can be seen as an approximation of what a single downstream neuron can extract from the latent representation and serves as a convenient evaluation measure. While in the cortex readouts may be more complex, i.e., consist of multiple processing stages, disentangled representations nevertheless are a convenient basis (e.g., Ha and Schmidhuber, 2018). We agree that a shift of focus towards representation learning is appropriate, in particular because semantization and memory consolidation in general, may include other aspects which are not captured by our current model. We have adapted the revised manuscript accordingly.

Miscellaneous comments.– Is it spelled semantization or semanticization? The latter appears to be in more common use.

Both are used, and it seems that memory-related articles tend to use semantization” (Dudai et al., 2015; Gilboa and Marlatte, 2017; Meeter and Murre, 2004; Sweegers and Talamini, 2014).

– I found the tSNE plots not particularly useful. tSNE is nonlinear so it is not a measure of linear category untangling. Please say more about what exactly this measure means, and report the perplexity parameter and how it was chosen.

We agree that tSNE visualization are not easily interpretable. In the revised manuscript, we thus use the PCA algorithm directly on *z* features (Figure 6).

– The authors should be aware of recent failures to replicate the Berkes (2011) result: https://elifesciences.org/articles/61942

We thank the reviewer for providing this reference. We have included it in the revised manuscript in section 3, lines 500-502.

Finally, some citations that I think could be mentioned:Previous proposals that the brain may learn adversarially:– Gershman, Samuel J. "The generative adversarial brain." Frontiers in Artificial Intelligence 2 (2019): 18.– https://arxiv.org/abs/2006.10811 (full disclosure, a work of my own)

We thank the reviewer for these references. In the revised manuscript, we include these in lines 417-420.

Work in the ML community in which the encoder is also a discriminator:– Brock, Andrew, Lim, Theodore, Ritchie, James M, and Weston, Nick. Neural photo editing with introspective adversarial networks. arXiv preprint arXiv:1609.07093, 2016.– Ulyanov, Dmitry, Vedaldi, Andrea, and Lempitsky, Victor. It takes (only) two: Adversarial generatorencoder networks. In Thirty–Second AAAI Conference on Artificial Intelligence, 2018.– Huang, Huaibo, He, Ran, Sun, Zhenan, Tan, Tieniu, et al. Introvae: Introspective variational autoencoders for photographic image synthesis. In Advances in neural information processing systems, pp. 52–63, 2018.– Munjal, Prateek, Paul, Akanksha, and Krishnan, Narayanan C. Implicit discriminator in variational autoencoder. arXiv preprint arXiv:1909.13062, 2019.– Bang, Duhyeon, Kang, Seoungyoon, and Shim, Hyunjung. Discriminator feature–based inference by recycling the discriminator of gans. International Journal of Computer Vision, pp. 1–23, 2020.

We thank the reviewer for these references. In the revised manuscript, these are included (section 2.1, lines 129-131).

References

Beckham, C., Honari, S., Verma, V., Lamb, A. M., Ghadiri, F., Hjelm, R. D., Bengio, Y., and Pal, C. (2019). On Adversarial Mixup Resynthesis. page 12.

Benjamin, A. S. and Kording, K. P. (2021). Learning to infer in recurrent biological networks.

Berthelot, D., Raffel, C., Roy, A., and Goodfellow, I. (2018). Understanding and Improving Interpolation in Autoencoders via an Adversarial Regularizer. *arXiv:1807.07543 [cs, stat]*. arXiv: 1807.07543.

Chen, T., Kornblith, S., Norouzi, M., and Hinton, G. (2020). A Simple Framework for Contrastive Learning of Visual Representations. *arXiv:2002.05709 [cs, stat]*. arXiv: 2002.05709.

DiCarlo, J. J., Zoccolan, D., and Rust, N. C. (2012). How Does the Brain Solve Visual Object Recognition? *Neuron*, 73(3):415–434.

Dudai, Y., Karni, A., and Born, J. (2015). The Consolidation and Transformation of Memory. *Neuron*, 88(1):20–32.

Dumoulin, V., Belghazi, I., Poole, B., Mastropietro, O., Lamb, A., Arjovsky, M., and Courville, A. (2017). Adversarially Learned Inference. *arXiv:1606.00704 [cs, stat]*. arXiv: 1606.00704.

Gershman, S. J. (2019). The Generative Adversarial Brain. *Frontiers in Artificial Intelligence*, 2.

Gilboa, A. and Marlatte, H. (2017). Neurobiology of Schemas and Schema-Mediated Memory. *Trends in Cognitive Sciences*, 21(8):618–631.

Goodfellow, I. J., Pouget-Abadie, J., Mirza, M., Xu, B., Warde-Farley, D., Ozair, S., Courville, A., and Bengio, Y. (2014). Generative adversarial networks.

Ha, D. and Schmidhuber, J. (2018). World models. *arXiv preprint arXiv:1803.10122*.

Hinton, G. E. (2012). A Practical Guide to Training Restricted Boltzmann Machines, pages 599–619. Springer Berlin Heidelberg, Berlin, Heidelberg.

Inostroza, M. and Born, J. (2013). Sleep for Preserving and Transforming Episodic Memory. *Annual Review of Neuroscience*, 36(1):79–102.

Jolliffe, I. T. and Cadima, J. (2016). Principal component analysis: a review and recent developments. Philosophical Transactions of the Royal Society A: Mathematical, Physical and Engineering Sciences, 374(2065):20150202.

Lewis, P. A., Knoblich, G., and Poe, G. (2018). How Memory Replay in Sleep Boosts Creative Problem-Solving. *Trends in Cognitive Sciences*, 22(6):491–503.

Lillicrap, T. P., Santoro, A., Marris, L., Akerman, C. J., and Hinton, G. (2020). Backpropagation and the brain. *Nature Reviews Neuroscience*.

Liu, X., Zhang, F., Hou, Z., Wang, Z., Mian, L., Zhang, J., and Tang, J. (2021). Self-supervised Learning: Generative or Contrastive. *arXiv:2006.08218 [cs, stat]*. arXiv: 2006.08218.

Mamelak, A. N. and Hobson, J. A. (1989). Dream Bizarreness as the Cognitive Correlate of Altered Neuronal Behavior in REM Sleep. *Journal of Cognitive Neuroscience*, 1(3):201–222.

Meeter, M. and Murre, J. M. J. (2004). Consolidation of Long-Term Memory: Evidence and

Alternatives. *Psychological Bulletin*, 130(6):843–857.

Nir, Y. and Tononi, G. (2010). Dreaming and the brain: from phenomenology to neurophysiology. *Trends in Cognitive Sciences*, 14(2):88–100.

Radford, A., Metz, L., and Chintala, S. (2015). Unsupervised Representation Learning with Deep Convolutional Generative Adversarial Networks. *arXiv:1511.06434 [cs]*. arXiv: 1511.06434.

Richards, B. A., Lillicrap, T. P., Beaudoin, P., Bengio, Y., Bogacz, R., Christensen, A., Clopath,

C., Costa, R. P., de Berker, A., Ganguli, S., Gillon, C. J., Hafner, D., Kepecs, A., Kriegeskorte, N., Latham, P., Lindsay, G. W., Miller, K. D., Naud, R., Pack, C. C., Poirazi, P., Roelfsema, P., Sacramento, J., Saxe, A., Scellier, B., Schapiro, A. C., Senn, W., Wayne, G., Yamins, D., Zenke, F., Zylberberg, J., Therien, D., and Kording, K. P. (2019). A deep learning framework for neuroscience. *Nature Neuroscience*, 22(11):1761–1770.

Simons, J. S., Garrison, J. R., and Johnson, M. K. (2017). Brain mechanisms of reality monitoring. *Trends in Cognitive Sciences*, 21(6):462–473.

Spano`, G., Pizzamiglio, G., McCormick, C., Clark, I. A., De Felice, S., Miller, T. D., Edgin, J. O., Rosenthal, C. R., and Maguire, E. A. (2020). Dreaming with hippocampal damage. *eLife*, 9.

Spoerer, C. J., McClure, P., and Kriegeskorte, N. (2017). Recurrent Convolutional Neural Networks: A Better Model of Biological Object Recognition. *Frontiers in Psychology*, 8.

Sweegers, C. C. and Talamini, L. M. (2014). Generalization from episodic memories across time: A route for semantic knowledge acquisition. *Cortex*, 59:49–61.

Tschannen, M., Djolonga, J., Rubenstein, P. K., Gelly, S., and Lucic, M. (2020). On mutual information maximization for representation learning.

Wattenberg, M., Vi´egas, F., and Johnson, I. (2016). How to use t-sne effectively. *Distill*.

Whittington, J. C. and Bogacz, R. (2019). Theories of Error Back-Propagation in the Brain. *Trends in Cognitive Sciences*, 23(3):235–250.

Wierzynski, C. M., Lubenov, E. V., Gu, M., and Siapas, A. G. (2009). State-dependent spike-timing relationships between hippocampal and prefrontal circuits during sleep. *Neuron*, 61(4):587–596.

Zbontar, J., Jing, L., Misra, I., LeCun, Y., and Deny, S. (2021). Barlow twins: Self-supervised learning via redundancy reduction. *arXiv preprint arXiv:2103.03230*.

[Editors’ note: what follows is the authors’ response to the second round of review.]

The manuscript has been improved and the reviewers were overall positive, but they have one significant remaining request and one minor request for a caveat:The significant request is to expand on how exactly the theory can be tested in experiments, with particular emphasis on diagnostic experiments that would allow us to rule out plausible alternatives. From Reviewer 3: "It takes a lot of thought to imagine how this particular hypothesis would surface in data and I don't think it should be left to the reader. More specifically, the paper still has no experimental predictions that could separate this idea from other similar possibilities involving generative models. The authors agreed in the response that they pose few predictions for the adversarial component, and instead only for "whether REM sleep is involved in cortical representation learning using a generative model." The anterior prefrontal cortex is now briefly mentioned as the top of the discriminator, but surely there would be a great deal of evidence in connectivity, power over plasticity, lesion studies, etc., that could confirm this. Re: "we interpret the reported novelty of REM dreams as strong existing evidence that this learning is based on adversarial principles rather than driven by reconstructions," (531-533): Here I disagree (though yes, not reconstructions). It is a strong hint that it is driven by an offline stage involving generative processes. This need not be adversarial. The Wake-Sleep algorithm, for example, also has a Sleep phase in which the hierarchical generative model generates samples via ancestral sampling from the top-level prior distribution. (Perhaps there is a misunderstanding regarding WS: the introduction currently dismisses the WS algorithm with the sentence, "these models explicitly try to reconstruct observed sensory inputs, while most dreams observed during REM sleep rarely reproduce past sensory experiences", lines 40-41. WS does try to reconstruct inputs during wake, but during sleep it 'fantasizes' randomly like a GAN.) Thus I still feel the paper does not offer tests by which we could know if the model were true."

The main request was to provide specific experimental predictions of our model, in particular predictions that distinguish the adversarial dreaming idea from other representation learning frameworks using generative models. Following these requests, we added a new section in the Discussion (”Signatures of adversarial learning in the brain”, lines 557-627). Here, we briefly summarize the main predictions.

First, we predict the existence of a cortical mechanism to distinguish externally from internally generated activity, in particular:

– That a specific subset of neurons implement the discriminator function; these neurons are characterized by being in opposite activity regimes during wakefulness and REM sleep.

– An inverse correlation between the veridicality of hallucinations and REM sleep quality in schizophrenic patients due to impaired discriminator function.

Second, our model makes predictions about plasticity on feedforward and feedback pathways, in particular:

– That the same low-level activity pattern triggers opposite plasticity during wakefulness and REM sleep on feedforward synapses due to the opposite target of the discriminator neurons.

– That plasticity occurs on both bottom-up and top-down pathways both during wakefulness and REM sleep; this is in strong contrast to alternative models involving offline states such as Waake-Sleep.

– sign-switch of the plasticity rule for synapses in feedback pathways between wakefulness and REM sleep, potentially testable by inferring plasticity rules from experimental observables.

The minor request is to acknowledge in the intro and the new discussion paragraph that the generative algorithm likely requires certain architectures & priors to deliver semantic representations.Typo: "Moreover, removing NREM from training also increases [this] ratio.".

The minor request was to acknowledge in the introduction and the discussion sections that the generative algorithm likely requires certain architectures and priors to deliver semantic representations. We modified the manuscript in lines 93-94 and 403-404 accordingly.